



Regional-scale paleofluid system across the Tuscan Nappe - Umbria Marche Arcuate Ridge (northern
Apennines) as revealed by mesostructural and isotopic analyses of stylolite-vein networks
Nicolas Beaudoin[1,*], Aurélie Labeur[1,2], Olivier Lacombe[2], Daniel Koehn[3], Andrea Billi[4], Guilhem
Hoareau[1], Adrian Boyce[5], Cédric M. John[7], Marta Marchegiano[7], Nick M. Roberts[6], Ian L. Millar[6], Fanny
Claverie[8], Christophe Pecheyran[8], and Jean-Paul Callot[1].
1. Universite de Pau et des Pays de l'Adour, E2S UPPA, LFCR, Pau, France
(nicolas.beaudoin@univ-pau.fr)
2. Sorbonne Université, CNRS-INSU, Institut des Sciences de la Terre de Paris - ISTeP, Paris,
France
3. GeoZentrum Nordbayern, University Erlangen-Nuremberg, Erlangen, Germany
4. Consiglio Nationale delle Ricerche, Roma, Italy
5. Scottish Universities Environmental Research Centre (SUERC), East Killbride, UK
6. Geochronology and Tracers Facility, British Geological Survey, Environmental Science Centre,
Nottingham, NG12 5GG, UK
7. Department of Earth Sciences & Engineering, Imperial College London, London, UK
8. Universite de Pau et des Pays de l'Adour, E2S UPPA, IPREM, Pau, France
Abstract
We report the results of a multi-proxy study that combines structural analysis of fracture-stylolite
network and isotopic characterization of calcite vein cements/fault coating. Together with new
paleopiezometric and radiometric constraints on burial evolution and deformation timing, these
results provide a first-order picture of the regional fluid pathways network during the main stages of
contraction in the Tuscan Nappe and Umbria Marche arcuate ridge (Northern Apennines).We
reconstruct four continuous steps of deformation at the scale of the belt: burial that developed
sedimentary stylolites, Apenninic-related layer parallel shortening with a contraction striking NE-SW,
local extension related to folding, then a late stage of fold tightening under a contraction still striking
NE-SW. In order to assess the timing and burial depth of strata at all stages, we combine a
paleopiezometric tool based on inversion of the roughness of sedimentary stylolites that constrains
the range of burial depth of strata prior to layer-parallel shortening, with burial models and U-Pb
absolute dating of fault coatings. In the western part of the ridge, layer-parallel shortening started in
Serravalian time (~12 Ma), then folding started at Tortonian time (~8 Ma), late stage fold tightening
started in early Zanclean (~5 Ma) and likely lasted until recent/modern extension occurred (~3 Ma
onward). This timing provides important constraints on the temperature that expectedly prevailed in
the studied strata through its history. The textural and geochemical ($\delta^{18}O$, $\delta^{13}C$, $\Delta_{47}CO_2$ and $^{87}Sr/^{86}Sr$)
study of calcite vein cements and fault coatings reveals that most of the fluids involved in the belt
during deformation are local, or flowed laterally from the same reservoir. However, the western edge
of the ridge recorded pulses of eastward squeegee-type migration of hydrothermal fluids (>140°C),
that can be related to the difference in structural style of the subsurface between the eastern Tuscan
Nappe and the Umbria Marche Ridge.
Introduction
The upper crust hosts ubiquitous fluid migrations that occur at all scales, leading to strain
localization, earthquake triggering and georesource generation, distribution and storage (e.g.,
Cartwright, 2007;Andresen, 2012;Bjørlykke, 1994, 1993;Lacombe and Rolland, 2016;Lacombe et al.,



2014;Roure et al., 2005;Agosta et al., 2016). Since an important part of the world's exploited
hydrocarbons, strategic ores and water resources are distributed in carbonate rocks (Agosta et al.,
2010), it is fundamental to be able to properly depict the history of fluid migration in deformed rocks,
not only to predict and monitor energy prospect and potential storage area, but also to understand
which mechanisms make fluids migrate in carbonate rocks, what are the time and spatial scales of
fluid flow involved, and what controls the diagenetic history of reservoirs.
Fluid migration events and related accumulations are usually linked to past tectonic events,
especially to the fault/fracture pattern created during these tectonic events. Indeed, structural studies
established that fracture networks in folded reservoirs are not exclusively related to the local folding
history (Stearns and Friedman, 1972) and can also witness burial history (Becker et al., 2010;Laubach
et al., 2010;Laubach et al., 2019) and long-term and large-scale regional deformation (Lacombe et al.,
2011;Quintà and Tavani, 2012;Tavani and Cifelli, 2010;Tavani et al., 2015a;Bellahsen et al.,
2006;Bergbauer and Pollard, 2004;Ahmadhadi et al., 2008;Sassi et al., 2012;Beaudoin et al., 2012). In
fold-and-thrust belts and orogenic forelands, it is for instance possible to subdivide the mesoscale
deformation (faults, veins, stylolites) history into specific stages: extension related to foreland flexure
and bulging; pre-folding layer-parallel shortening (kinematically unrelated with folding); early folding
layer-parallel shortening; syn-folding, strata curvature-related, local extension; late stage fold
tightening, the last three stages being kinematically related with folding; and post folding contraction
or extension (kinematically unrelated with folding). In the past decades, a significant volume of work
has thus been conducted in order to reconstruct past fluid migrations through either localized fault
systems or distributed sub-seismic fracture networks, in relationship with past tectonic events from
the scale of a single fold to that of the basin itself (Engelder, 1984;Reynolds and Lister, 1987;McCaig,
1988;Evans et al., 2010;Forster and Evans, 1991;Cruset et al., 2018;Lacroix et al., 2011;Travé et al.,
2000;Travé et al., 2007;Bjørlykke, 2010;Callot et al., 2017a;Callot et al., 2017b;Roure et al., 2010;Roure
et al., 2005;Van Geet et al., 2002;Vandeginste et al., 2012;Vilasi et al., 2009;Barbier et al.,
2012;Beaudoin et al., 2011;Beaudoin et al., 2014;Beaudoin et al., 2013). Studies highlighted that large-
scale faults and sub-seismic scale fracture networks alike can impact the local fluid system, connecting
compartments vertically and leading to local invasion of distant, hydrothermal fluids, over different
time scales at the fold-scale (Beaudoin et al., 2011;Evans and Hobbs, 2003;Evans and Fischer,
2012;Barbier et al., 2012;Fischer et al., 2009;Lefticariu et al., 2005;Di Naccio et al., 2005). Fracture
networks and related mineralizations can also be successfully used to describe the fluid system at the
regional scale, with long term across-strike, stratigraphically-compartmentalized, fluid migration
directed by compressive tectonic stress, with in some case an opening to external fluid flow, such as
downward migration of meteoric fluids, or upward migration of hydrothermal fluids (*i.e.* hotter than
the host-rock they precipitated in) of various origins (meteoric, marine, metamorphic), (Roure et al.,
2005;Vandeginste et al., 2012;Cruset et al., 2018;Lacroix et al., 2011;Travé et al., 2000;Travé et al.,
2007;De Graaf et al., 2019;Callot et al., 2010;Beaudoin et al., 2014;Bertotti et al., 2017;Gonzalez et al.,
2013;Lucca et al., 2019;Mozafari et al., 2019;Storti et al., 2018;Vannucchi et al., 2010).
This contribution reports an orogen-scale paleofluid flow study in the Northern Apennine (Italy).
The study builds upon the mesostructural and geochemical analysis of vein and stylolite networks
within the competent Jurassic-Oligocene carbonate platform along a transect running across the
Tuscan Nappe (TN) and the Umbria-Marche Apennines Ridge (UMAR) (Fig. 1a). The data collection was
organized to cover a large area comprising several folds in order to be able to differentiate regional
trends from local, fold-related ones. We focused on identifying and characterizing the first order



pattern of mesostructures – faults, fractures and stylolites – associated with LPS and with thrust-
related folding, along with the stable isotope signatures ($\delta^{18}O$, $\delta^{13}C$), radiogenic signatures ($^{87}Sr/^{86}Sr$),
clumped isotope signature ($\Delta_{47}CO_2$), and U-Pb absolute dating of their calcite cements. Without an
appraisal of which fracture trends are relevant to the large scale (*i.e.*, regional) tectonic evolution,
there was a risk to otherwise capture mesostructural and geochemical signals of local meaning only.
In order to discuss the local versus hydrothermal fluid origin, we also considered burial curves derived
from published dataset coupling sedimentary data and organic matter paleothermometers. Novel
constraints are added to the timing and minimal depth of LPS-related deformation based on the study
of the roughness properties of bedding-parallel stylolites, the inversion of which reliably returns the
maximum depth at which compaction under a vertical maximum principal stress was still prevailing in
the strata. U-Pb absolute dating of calcite steps on mesoscale faults further constrains the timing of
folding. Such a multi-proxy approach, that combines structural analysis of fracture-stylolite network
and isotopic characterization of cements, together with new constraints on burial evolution and
deformation timing, provides for the first time a picture of the regional fluid pathways during the main
stages of the Apenninic contraction.

1.   Geological setting
The Neogene-to-Quaternary Apennines fold-and-thrust belt results from the convergence of
Eurasia and Africa (Lavecchia, 1988;Elter et al., 2012). It is associated with the eastward retreating
subduction of the Adriatic Plate under the European plate. The Apennines extend from the Po Plain
to the Calabrian arc, and are divided into two main arcs, the Northern Apennines that extend down to
the south of the UMAR, and the Southern Apennines that cover the remaining area down to the
Calabrian arc (Carminati et al., 2010). The evolution of the Apennines is characterized by a roughly
eastward migration of thrust fronts and associated foredeep basins, superimposed by post-orogenic
extension at the rear of the eastward propagating orogenic belt (Cello et al., 1997;Tavani et al.,
2012;Lavecchia, 1988;Ghisetti and Vezzani, 2002).
The study area, the Tuscan Nappe and the Umbria-Marches Apennines Ridge, comprises a
succession of carbonate rocks, Late Triassic to Oligocene in age, which corresponds to a carbonate
platform (Lavecchia, 1988;Carminati et al., 2010). The Umbrian carbonate units overlie early Triassic
evaporites that act as a décollement level, itself unconformably overlying the crystalline basement
rocks (Fig. 1b). Above the platform, Miocene turbidite deposits witness the progressive eastward
involvement of the platform into the fold-and-thrust belt (Calamita et al., 1994). In the western part
of the area, the belt is a thin-skinned assembly of piggy back duplex folds (Fig. 1c), the so-called Tuscan
Nappe (TN), the folding and thrusting of which started by the Late Aquitanian and lasted until the
Langhian (Carboni et al., 2020). The UMAR is an arcuate ridge exhibiting an eastward convexity, with
a line connecting Perugia and Ancona separating a northern part where structural trends are oriented
NW-SE, from a southern part where structure trends are oriented N-S (Calamita and Deiana, 1988).
Burial models suggest that, from Burdigalian to early Messinian times, the TN was further buried under
the allochthonous Ligurian thrust sheet, reaching locally up to 1 km in thickness (Caricchi et al., 2015).
In the eastern part (now UMAR), the foreland was progressively folded and thrusted from the Lower
Miocene in the westernmost part of the current ridge to the Messinian in the foreland of the ridge
(Mazzoli et al., 2002). UMAR has been considered for long as a thin-skinned thrust belt where
shortening was accommodated by stacking and duplexing of sedimentary units detached above a
décollement level located in the Triassic evaporites (Conti and Gelmini, 1994;Carboni et al., 2020). The



seismic profile of the CROsta Profonda (CROP) project led authors to interpret the UMAR as resulting
from thick-skinned tectonics, where the basement is involved in shortening (Barchi et al., 1998)
through the positive inversion of normal faults inherited from the Jurassic Tethyan rifting (Fig. 1c).
Even if the implication of the basement in shortening is seemingly more accepted now, the subsurface
geometry is still debated, with some models involving shallow duplexes (Tavarnelli et al.,
2004;Mirabella et al., 2008), while in more recent works surface folds are rather interpreted as related
to high angle thrusts that either sole within the mid-Triassic décollement level, or involve the
basement (Scisciani et al., 2014;Scisciani et al., 2019;Butler et al., 2004)(Fig. 1c). In these last views,
the style of deformation of the UMAR strongly contrasts with the style of deformation of the TN where
shortening is accommodated by allochtonous, far-travelled duplex nappes (Carboni et al., 2020)(Fig.
1c). The cross-section of Figure 1c also implies that at least part of the motion on the décollement
level at the base of the TN postdates the westernmost activation of steep thrusts of the UMAR, as the
thrust at the base of the TN cuts and offsets the west-verging basement fault in the area of Monte
Subasio. Nowadays, the whole TN-UMAR area undergoes extension, with numerous active normal
faults developing trenches, as the contraction front migrated toward the Adriatic Sea (d'Agostino et
al., 2001).
Our sampling focused on the carbonate formations cropping out from W to E in the Cetona area
located west from Perugia; the Monte Corona in the TN; the Monte Subasio, Gubbio Area, Spoletto
Area, Monte Nero, Monte San Vicino, and Monte Cingoli in the UMAR, and the Monte Conero, the
youngest onshore anticline related to the Apenninic compression, located on the coast line (Fig. 1a).
The sampled units comprise, following the stratigraphic order (Fig. 1b): the Triassic anhydrites and
dolostones of the Anidridi di Burano Formation with limestone and marl intercalation at the top; (2)
Liassic massive dolomites of the Calcare Massiccio Fm. (Hettangian to Sinemurian); (3) the grey
Jurassic limestones with chert beds of the Corniola Fm. (Lothangian-Pleisbachian); (4) the micritic
limestones, marls, and cherts of the Bosso/Calcare Diasprini Fm (Toarcian-Tithonian); (5) the white
limestones with chert beds of the Maiolica Fm. (Tithonian-Aptian); (6) the marly limestones of the
Fucoidi Fm. (Aptian-Cenomanian); (7) the white marly limestones of the Scaglia Bianca Fm.
(Cenomanian); (8) the pink marly limestones of the Scaglia Rossa Fm. (Turonian-Priabonian); and (9)
the grey marly limestones of the Scaglia Cinera Fm. (Priabonian- Cattian). Up to 3000m of Miocene
turbidites were deposited when the area of interest was the foredeep ahead of the advancing fold-
and-thrust belt and during fold development, including clay-rich limestones and silts of Marnoso-
Aranacea (Aquitanian-Tortonian); in the eastern part of the ridge (east from the Cingoli anticline),
thicker foredeep deposits are Messinian to Pliocene in age.

2.  Methods and results
We used structural and geochemical method to characterize the scenario of fluid rock interaction
during deformation of the Umbria-Marche arcuate ridge in the Northern Apennines. Below, for each
method, we explain the method itself and then we report the related results. We favour the
presentation of the methods and related results in closed succession to make the latter as
comprehensible as possible.
a.  Mesostructural analysis of joints, veins and striated fault planes
i.  Methodology



~1300 joint and vein orientations, along with tectonic stylolite orientations, were measured
along a WSW-ENE transect going from Cetona in the TN to Monte Conero on the coastline (Fig. 1a).
For each measurement site, fractures and tectonic stylolites (*i.e.* bedding perpendicular dissolution
planes displaying horizontal peaks after unfolding or vertical dissolution planes displaying horizontal
peaks in the current bed attitude) were measured. Chronological relationships were carefully
observed in the field (Fig. 2) and checked in thin sections under the optical microscope when possible
(Fig. 3). It is worthwhile noting that the veins of sets J1 and J2 show twinned calcite grains (Fig 3) with
mostly thin and rectilinear twins (thickness < 5 μm)(Fig. 3). Poles to fractures and stylolite peaks were
projected on Schmidt stereograms, lower hemisphere, in the current attitude of the strata (Raw), and
after unfolding (Unfolded) as well (Fig. 4). Assuming the same mode of deformation (*i.e.* mode I
opening joints/veins) and consistent chronological relationships and orientation, we use Fisher density
to define statistically meaningful sets of joints. Tectonic stylolite planes and peaks were measured,
and we consider that the average orientation of the stylolite peaks at the fold scale represents the
orientation of the horizontal maximum principal stress σ1, as peaks grow parallel to the main
shortening direction. To complement this mesostructural analysis, striated fault planes were
measured (1) in the Langhian carbonates from the syncline West from San Vicino, and (2) in the
forelimb of the Monte Subasio, with one site in the Scaglia Cinera and one site in the Scaglia Rossa e
Bianca. At each site, paleostress orientations (local trend and plunge) and regimes were calculated
using inversion techniques (Angelier, 1984). Published studies in the UMAR highlight the complexity
of fracture patterns at the fold scale, that witness several phases of stress perturbation and
stress/block rotation due to the local tectonics and structural inheritance(Tavani et al.,
2008;Petracchini et al., 2012;Beaudoin et al., 2016;Díaz General et al., 2015). In order to capture the
mesostructural and fluid flow evolution at the regional scale during layer-parallel shortening and
folding, we gathered the most representative fracture data by structure, regardless of the structural
complexity in the individual folds, and corrected them from the local bedding dip to discriminate
between early and syn-folding features.
ii.   Results
Based on the average orientation and the angle to the local fold axis, veins/joints can be
gathered in 2 sets labelled J (Fig. 4): a first set J1 gathers joints/veins at high angle to bedding, that
strike E-W to NE-SW but perpendicular to the local strike of fold axis. The trend of this set J1 evolves
eastward as follows: E-W in the westernmost part (Cetona, Subasio), E-W to NE-SW in the central part
(Catria, Nero), NE-SW in the eastern part of the chain (San Vicino, Cingoli), and ENE-WSW in the far
foreland (Conero). The second set J2 gathers joints/veins at high angle to bedding that strike parallel
to the local trend of the fold hinge, *i.e.* NW-SE in the ridge to N-S in the outermost part of the belt,
where the arcuate shape is more marked. Note that as set J1 strikes perpendicular to the local strata
direction, it is impossible to infer a pre-tilting or post-tilting (then called J3 hereinafter) origin for its
development. In most case though, abutment relationships establish a relative chronology with set J1
predating set J2 (Fig. 3). Also, a third set comprising joints striking N-S while oblique to the direction
of the fold axis is documented in the Monte Catria, but will not be considered further as it is not
encountered elsewhere in the chain. Tectonic stylolites can be gathered as sets labelled S based on
the orientation of their peaks. At first order, stylolites of which peaks are oriented NE-SW prevail :
they are either bed-perpendicular, vertical with horizontal peaks in the unfolded attitude of strata,
thus predating tilting (set S1), or ~vertical with ~horizontal peaks in the current attitude of strata, thus
postdating tilting (set S2). However, because (1) stylolite data were often collected in shallow dipping





strata, (2) peaks are not always perpendicular to the stylolite planes and (3) the orientation data are
scattered with intermediate plunges of the peaks, S1 and S2 are not always easily distinguished when
both occurred. Another set showing stylolite planes with N-S peaks parallel to bedding, thus predating
folding, is documented only at Monte Subasio, thus will not be integrated in the sequence at the scale
of the fold-and-thrust belt. Finally, some mesoscale reverse and strike-slip conjugate fault systems
have been measured (sets labelled F), of which fault-slip data inversion under specific assumptions
(e.g., Lacombe, 2012) yields (1) a NE-SW contraction in the unfolded attitude of the strata (early
folding set F1, bedding-parallel faults) and (2) a NE-SW contraction in the current attitude of the strata
(late folding set F2, strike-slip conjugate faults and reverse faults).

b.   Inversion of sedimentary stylolites
i.   Methodology

Bedding-parallel stylolites are rough dissolution surfaces that developed in carbonates in flat
laying strata during burial at the time when $\sigma_1$ was vertical. As proposed by Schmittbuhl et al. (2004)
and later developed by Koehn et al. (2012), Ebner et al. (2009b);Ebner et al. (2010), Rolland et al.
(2014) and Beaudoin et al. (2019);Beaudoin et al. (2020), the 1-D roughness of a track along the
bedding-parallel stylolite (*i.e.* difference in height between two points along the track) results from a
competition between roughening forces (*i.e.* pining on non-soluble particles in the rocks) and
smoothening forces (*i.e.* the surface energy at scale typically < 1mm, and the elastic energy at scale >
1mm). The stylolite growth model (Koehn et al., 2007;Ebner et al., 2009a;Rolland et al., 2012;Toussaint
et al., 2018) predicts that surface energy-controlled scale returns a steep slope characterized by a
roughness exponent (so-called Hurst exponent) of 1.1, while the elastic energy-controlled scale
returns a gentle slope with a roughness exponent of 0.6 (Fig. 5). The length at which the change in
roughness exponent occurs, called the cross-over length (Lc, in mm), is directly related to the
magnitude of differential and mean stress ($\sigma_d = \sigma_1 - \sigma_3$ and $\sigma_m = \frac{\sigma_1+\sigma_2+\sigma_3}{3}$, respectively, in Pa)
prevailing in the strata at the time the stylolite stopped to be an active dissolution surface following:
$$Lc = \frac{\gamma E}{\beta \sigma_m \sigma_d} \qquad (1)$$
where E is the Young modulus of the rock (in Pa), γ is the solid-fluid interfacial energy (in J.m⁻²), and
β= ν(1-2ν)/π, a dimensionless constant with ν being the Poisson ratio. Samples of bedding parallel
stylolites of which peaks were perpendicular to the dissolution plane were collected in specific points
of the study area, and several stylolites were inverted. The inversion process follows the method
described in Ebner et al. (2009b). Samples were cut perpendicular to the stylolite, hand polished to
enhance the visibility of the track while being cautious about not altering the peaks, scanned at high-
resolution (12800 pixel per inchs), and the 1D track was hand drawn with a pixel-based software
(GIMP). Each track was analyzed as a periodic signal by using the Average Wavelet Spectrum with
Daubechies D4 wavelets (Fig. 5) (Ebner et al., 2009b;Simonsen et al., 1998). In the case of bedding
parallel stylolite related to compaction and burial, we assume the horizontal stress is isotropic in all
direction ($\sigma_v$>>$\sigma_h$=$\sigma_H$) to simplify the equation 1 (Schmittbuhl et al., 2004) as:
$$\sigma_v^2 = \frac{\gamma E}{\alpha Lc} \qquad (2)$$
where $\alpha = \frac{(1-2\nu)*(1+\nu)^2}{30\pi(1-\nu)^2}$. According to the sampled formation, we used the solid-fluid interfacial
energy γ of 0.24 J.m⁻² for dolomite, and of 0.32 J.m⁻² for calcite (Wright et al., 2001). As an





approximation for the material mechanical properties, we use a classic Poisson ratio of ν=0.25 ±0.05,
and the average Young modulus derived from the Jurassic-Eocene competent core of E=24.2 GPa
(Beaudoin et al., 2016). It is important to note that because of the non-linear regression method we
use, and because of uncertainty on the mechanical parameters of the rock at the time it dissolved, the
uncertainty on the stress has been calculated to be about 12% (Rolland et al., 2014). As the dissolution
occurs along a fluidic film (Koehn et al., 2012;Rolland et al., 2012;Toussaint et al., 2018), the stylolite
roughness is unaffected by local fluid overpressure until the system is fluidized and hydrofractures
(Vass et al., 2014), meaning it is possible to translate vertical stress magnitude directly into depth if
considering an average dry rock density for clastic/carbonated sediments (2400 g.m$^{-3}$, Manger (1963)),
without any additional assumption on the past thermal gradient or fluid pressure (Beaudoin and
Lacombe, 2018). This technique has already provided meaningful results in various settings (Bertotti
et al., 2017;Rolland et al., 2014;Beaudoin et al., 2019;Beaudoin et al., 2020).

ii.  Results

The paleopiezometric study of 30 bedding-parallel stylolites returned a range of burial depths,
across the UMAR, from W to E, reported in table 1. Most data come from the western part of the
UMAR: in the Subasio Anticline (n=7), the depth returned by the Scaglia Bianca and the lower part of
the Scaglia Rossa Fms. ranges from ca. 800 ± 100 m to ca. 1450 ± 150 m. In Fiastra area (n=6), the
depth returned for the Maiolica Fm. ranges from 800 ± 100 m to 1200 ± 150 m. In the Gubbio fault
area (n=4), the depth returned for the Jurassic Corniola Fm. ranging from 600 ± 70 m to 1450 ± 150 m.
In the Monte Nero (n=11), the depth data published by Beaudoin et al., (2016), and updated here
range from 750 ± 100 m to 1350 ± 150 m in the Maiolica. Fewer data comes from the western part of
the UMAR: in the Monte San Vicino (n=2), the depth returned for the Maiolica Fm. ranges from 1000
± 100 m to 1050 ± 100 m. Finally, the depth reconstructed for the lower part of the Scaglia Rossa is
650 ± 70 m in the foreland at Conero Anticline (n=1).

c.  Isotopic characterization of paleofluids (1) : O, C stable isotopes
i.  Methodology

Calcite cements that filled up tectonic veins related either to layer-parallel shortening or to
strata curvature at fold hinges were studied petrographically (Fig. 3). The vein textures were
characterized in thin sections under an optical microscope, and diagenetic states were checked under
cathodoluminescence, using a cathodoluminescence CITL CCL 8200 Mk4 operating under constant gun
condition of 15kV and 300μA. To perform Oxygen and Carbon stable isotope analysis on the cements
that were the most likely to witness the conditions of fluid precipitation at the time the veins opened,
we selected those veins that (1) show no obvious evidence of shear; (2) the texture of which was
elongated blocky or fibrous (Fig. 3); and (3) show homogeneous cement under cathodoluminescence
(Fig. 3), precluding any posterior diagenetic alteration.

40 μg of calcite powder was hand sampled for each of 58 veins and 54 corresponding host-
rocks in various structures and formation along the transect, in both TN and UMAR. Carbon and
oxygen stable isotopes were analyzed at the Scottish Universities Environmental Research Centre
(SUERC, East Kilbride, UK) on an Analytical Precision AP2003 mass spectrometer equipped with a
separate acid injector system. As samples were either pure calcite or pure dolomite, we placed
samples in glass vials to conduct a reaction with 105% H3PO4 under a helium atmosphere at 90°C.
Results are reported in table 2, in permil relative to Vienna PeeDee Belemnite (‰ VPDB). Mean





analytical reproducibility based on replicates of the SUERC laboratory standard MAB-2 (Carrara
Marble) was around ± 0.2‰ for both carbon and oxygen. MAB-2 is an internal standard extracted
from the same Carrara Marble quarry, as is the IAEA-CO208 1 international standard. It is calibrated
against IAEA-CO-1 and NBS-19.

ii.    Results

At the scale of the study area, most formations cropping out were sampled (Table 2), and
oxygen isotopic signatures of the vein cements and striated fault coatings range from -16.8‰ to 3.7‰
PDB while in the host rocks values range from -5.28 ‰ to 0.4‰ PDB. Carbon isotopic signatures range
from -9.7‰ to 2.7‰ PDB, and from 0‰ to 3.5‰ PDB in the veins and in the host rock, respectively
(Fig. 6a-b). Isotopic signatures are represented either according to the structure where they have been
sampled, irrespective of the structural position in the structure (*i.e.* limbs or hinge), or according to
the set they belong to, differentiating the sets J1, J2 and F1 (Fig. 6c). At the scale of the belt, isotopic
signatures of host rocks are very similar, the only noteworthy point being that the Triassic carbonates
are more depleted than the rest of the column ($\delta^{18}$O of -5.5‰ to -3.5‰ versus -3.2‰ to -1‰ PDB ).
Considering the vein cements, an isotopic trend arises in the Jurassic-Eocene rocks with more depleted
$\delta^{18}$O values in or near the Tuscan nappe than in the UMAR (Fig. 6d), completely irrespective of the
vein set, hence of the timing of opening. Especially, Monte Subasio and Monte Corona exhibit veins
with very depleted $\delta^{18}$O signatures < -15‰ PDB, while the most depleted $\delta^{18}$O value in the UMAR is -
7.3‰ PDB in the Monte San Vicino (Table 2). For the same dataset, the $\delta^{13}$C values are rather similar
in all structures and in all sets, a vast majority of veins showing cements with signatures of 1.5± 1.5‰
PDB.

321          d.    Isotopic characterization of paleofluids (2): Sr radiogenic isotopes
i.    Methodology

$^{87}$Sr/$^{86}$Sr analysis was performed at the BGS in Keyworth (UK) with a VG Sector 54-30 multiple
collector thermal ionization mass spectrometer. Mg-samples were loaded onto single Re filaments
with a Ta-activator. An $^{88}$Sr intensity of ~1 × 1011 A ± 10% was maintained. $^{87}$Sr/$^{86}$Sr was corrected for
mass fractionation to $^{86}$Sr/$^{88}$Sr = 0.1194 after the exponential law described in Nier (1938). The 2
standard error internal precision on individual analyses ranges between 12 and 17 ppm (smaller than
external reproducibility).

ii.    Results

Analyses were carried out on 7 veins and 6 corresponding host rocks, spread on three
structures of the UMAR (Monte Subasio, Monte Nero and Monte San Vicino, from the hinterland to
the foreland) and three formations (the Calcare Massiccio, the Maiolica, and the Scaglia Fms., Fig. 7,
Table 2). Vein sets sampled are the J1, J2 and J3 sets described in the whole area. Radiogenic
signatures of host rocks spread in three different sets, one more radiogenic (Scaglia Rossa,$^{87}$Sr/$^{86}$Sr ≈
0.7078) a second one less radiogenic (Calcare Massicio, $^{87}$Sr$^{/86}$Sr ≈0.7076), and a third one even less
radiogenic (Scaglia Bianca and Calcare Rupestre $^{87}$Sr$^{/86}$Sr ≈ 0.7073). Radiogenic signatures of host
rocks are in line with expected values for seawater at the time of their respective deposition (McArthur
et al., 2001). The radiogenic signatures of veins scatter from 0.7074 to 0.7077, with less radiogenic
values in the Monte Nero and in the Monte San Vicino (sets J1, J2), and more radiogenic signatures in
the Monte Subasio (Set J2). One vein cement of J3 in the Monte Subasio returned a lower radiogenic
value of 0.7074.





e. Isotopic characterization of paleofluids (3): Carbonate clumped isotope
paleothermometry ($\Delta_{47}\,CO_2$)
i. Methodology
Clumped isotopes analyses were carried out in the Qatar Stable Isotope Laboratory at Imperial
College London. The technique relies on the tendency for heavy isotopes ($^{13}C$, $^{18}O$) to 'clump' together
in the same carbonate molecule, that varies only by temperature. Since the clumping of heavy
isotopes within a molecule is a purely stochastic process at high temperature but is systematically
over-represented (relative to randomly distributing isotopes among molecules) at low temperature,
the 'absolute' temperature of carbonate precipitation can be constrained using clumped isotope
abundances.
Typical sample size was 3.5 mg of carbonate powder per replicate. Measurement of $^{13}C$-$^{18}O$
ordering in sample carbonate is achieved by measurement of the relative abundance of the $^{13}C^{18}O^{16}O$
isotopologues (mass 47) in acid evolved $CO_2$ and is referred in this paper as $\Delta_{47}\,CO_2$. Samples were
prepared on the automated clumped isotope measurement system (the IBEX: Imperial Batch
Extraction system): the IBEX was developed at Imperial College London and is manufactured and
distributed by Protiumms. A single run on the IBEX comprises 40 analysis, 30% of which are standards.
Each analysis takes about 2 hours. The process starts with 10 minutes of reaction of the carbonate
powder in a common acid bath containing 105% orthophosphoric acid at 90˚C to liberate $CO_2$. The $CO_2$
gas is then captured in a water/$CO_2$ trap maintained at liquid nitrogen temperature, and then moved
through a hydrocarbon trap filled with poropak and a second water trap using helium as carrier gas.
At the end of the cleaning process, the gas is transferred into a cold finger attached to the mass
spectrometer, and into the bellows of the mass spectrometer. Following transfer, analyte $CO_2$ was
measured on a dual inlet Thermo MAT 253 mass spectrometers (MS "Pinta"). The reference gas used
is a high purity $CO_2$, with the following reference values: -37.07‰ $\delta^{13}C_{VPDB}$, 8.9‰ $\delta^{18}O_{VSMOW}$.
Measurements comprise 8 acquisitions each with 7 cycles with 26s integration time. A typical
acquisition time is 20 minutes, corresponding to a total analysis time of 2 hours.
Data processing was carried out in the freely available stable isotope management software,
"Easotope" (John and Bowen, 2016) (www.easotope.org). The raw $\Delta_{47}\,CO_2$ is corrected in three steps:
mass spectrometer non-linearity was corrected by applying a "pressure baseline correction"
(Bernasconi et al., 2013). Next, the $\Delta_{47}$ results were projected in the absolute reference frame or
Carbon Dioxide Equilibrated Scale (CDES, Dennis et al. (2011)) based on routinely measured ETH1,
ETH2, ETH3, ETH4 and Carrara Marble (ICM) carbonate standards (Meckler et al., 2014;Muller et al.,
2017). The last correction to the raw $\Delta_{47}$ was to add an acid correction factor of 0.082‰ to obtain a
final $\Delta_{47}CO_2$ value (Defliese et al., 2015). Temperatures of precipitation can then be estimated using
the equation of Davies and John (2019). The bulk isotopic value of $\delta^{18}O$ is corrected for acid digestion
at 90°C by multiplying the value by 1.0081 using the published fractionation factor (Kim et al., 2007).
Contamination was monitored by observing the values on mass 48 and 49 from each measurement,
using a $\Delta_{48}$ offset value > 0.5‰ and/or a 49 parameter values > 0.3 as a threshold to exclude individual
replicates from the analysis (Davies and John, 2019).
ii. Results
13 samples were analyzed (Table 3), including cements of NE-SW (J1) and NW-SE (J2) pre-
folding vein sets, along with coatings of early folding reverse (F1) and late folding strike-slip conjugate
mesoscale faults (F2). Regardless of the structural position in the individual folds, veins were sampled





385 in the Monte Corona (TN), and in the UMAR at the Monte Subasio, the Monte San Vicino and the
386 syncline to its west. Analysis of $\Delta_{47}CO_2$ returns the precipitation temperature (T) and the oxygen
387 isotopic signature of the mineralizing fluid can be calculated using the $\delta^{18}O$ of the mineral, the clumped
388 isotope temperature and the equation of Kim et al. (2007) (Fig. 8). Veins and faults belong to the
389 Calcare Massiccio Fm., the Maiolica Fm., the Scaglia Fm., and the marls of the Langhian (Table 3). In
390 the outermost structure studied (Monte Corona), the fractures of set J2 (n=2) return consistent
391 precipitation temperatures T= 106 ± 8°C and $\delta^{18}O_{fluids}$ = 0 ± 1.8‰ VSMOW; the sample of the set J1
392 yields a T= 56 ± 16 °C and $\delta^{18}O_{fluids}$ = -1.1 ± 1.8‰ VSMOW; in the UMAR, at the Subasio anticline, set
393 F1 (n=3) returns temperatures T ranging from 80±5°C to 141±19°C and a corresponding $\delta^{18}O_{fluids}$
394 ranging from 8.4±1‰ to 16.1±2.1‰ VSMOW, while the set J2 returns a T=71±0°C and $\delta^{18}O_{fluids}$ = -
395 5.2±0‰ VSMOW; in the Monte Nero, set J1 (n=2) yields consistent T = 30±15°C and $\delta^{18}O_{fluids}$ =[ 2.7±2.4
396 to 6.8±0.2]‰ VSMOW; in the syncline on the west of the Monte San Vicino, set F2 (n=2) return
397 T=[36±4 to 70±7]°C and $\delta^{18}O_{fluids}$ =[ 2.5±0.7 to 8.3±1.2]‰ VSMOW; in the Monte San Vicino, set J1
398 yields a T = 47 ± 5°C and $\delta^{18}O_{fluids}$ = 3.0±1.1‰ VSMOW while set J2 yields a T= 74± 10°C and $\delta^{18}O_{fluids}$ =
399 7.2±1.6‰ VSMOW.

401    f. U-Pb absolute dating of veins and faults
402      i. Methodology

403 The Calcite U-Pb geochronology was conducted in two different ways, of which specific methodology
404 is reported as supplementary material:

405 - LA-ICPMS trace elements and U-Pb isotope mapping were performed at the Geochronology and
406 Tracers Facility, British Geological Survey, UK, on 6 veins samples. Data were generated using a Nu
407 Instruments Attom single collector inductively coupled plasma mass spectrometer coupled to a
408 NWR193UC laser ablation system fitted with a TV2 cell, following protocol reported previously
409 (Roberts et al., 2017;Roberts and Walker, 2016).

410 - LA-ICPMS U-Pb isotope mapping approach was undertaken at the Institut des Sciences Analytiques
411 et de Physico-Chimie pour l'Environnement et les Matériaux (IPREM) Laboratory (Pau, France). All the
412 29 samples were analysed with a 257 nm femtosecond laser ablation system (Lambda3, Nexeya,
413 Bordeaux, France) coupled to an HR-ICPMS Element XR (ThermoFisher Scientific, Bremen, Germany)
414 fitted with the Jet Interface (Donard et al., 2015). The method is based on the construction of isotopic
415 maps of the elements of interest for dating (U,Pb,Th) from ablation along lines, with ages calculated
416 from the pixel values (Hoareau et al., 2020). The ablation was made in a helium atmosphere (600 mL
417 $min^{-1}$), and 10 mL $min^{-1}$ of nitrogen was added to the helium flow before mixing with argon in the
418 ICPMS. Measured wash out time of the ablation cell was ~500 ms for helium gas. The fs-LA-ICP-MS
419 coupling was tuned on a daily basis, and the additional Ar carrier gas flow rate, torch position and
420 power were adjusted so that the U/Th ratio was close to 1 +/- 0.05 when ablating the glass SRM
421 NIST612. Detector cross-calibration and mass bias calibration were checked daily. The laser and HR-
422 ICPMS parameters used for U-Pb dating are detailed in the supplementary material.


424      ii. Results

425 Of the 35 samples screened for favorable U-Pb ratios, only 2 were selected for U-Pb dating (FAB5
426 and FAB6). Noteworthy, all samples from veins, whatever the set they belong to, reveal to have a U/Pb



ratio not high enough to return an age, with a too low U content and/or dominated by common lead
(see Supplementary material), which seems to be common in tectonic veins (Roberts et al., 2020).
Only two samples consisting of calcite fault coating provided suitable material and were further
analysed. Among these two, one could be successfully dated (FAB5). Despite a large majority of pixel
values dominated by common lead with some scatter, the pixels with higher U/Pb ratios made it
possible to obtain identical ages within the limits of uncertainty for the different plots (5.03 ± 1.2 Ma,
4.92 ± 1.3 Ma and 5.28 ± 0.95 Ma for the TW, the 86TW and the isochron plot, respectively) (Fig 9a).
The rather large age uncertainties are consistent with the moderately high RSE values, but the d-
MSWD values close to 1 indicate good alignment of discretized data (Fig. 9B). The other sample (FAB6)
gave distinct ages according to the plot considered, ranging from 2.17 ± 1.4 Ma to 6.53 ± 2 Ma, due to
low U/Pb ratios. Keeping in mind their low reliability, the ages obtained for this sample grossly point
toward precipitation younger than ~8 Ma.

3.  Interpretation of results and discussion
a.  Sequence of fracturing events and related regional compressional and extensional
trends

The previously defined joint/vein, fault and stylolite sets were compared and gathered in order to
reconstruct the deformation history at the scale of the belt. We interpret the mesostructural network
as witnessing three stages of regional deformation, supported by published fold-scale fracture
sequence ((Tavani and Cifelli, 2010;Tavani et al., 2008;Petracchini et al., 2012;Beaudoin et al.,
2016;Díaz General et al., 2015;Di Naccio et al., 2005;Vignaroli et al., 2013), in line with the ones
observed in most recent studies (see Evans and Fischer, 2012; Tavani et al., 2015a for reviews) *:*
*Layer parallel shortening (LPS)* stage: chronological relationships statistically suggest that set J1
formed before set J2. Set J1 is kinematically consistent with set S1 that recorded the NE-SW Apenninic
contraction, except in some places where sets J1 and S1 rather formed under a slight local
rotation/perturbations of the NE-SW compression as a result of structural inheritance and/or of the
arcuate shape of the fold. Bedding-parallel reverse faults of set F1 also belong to this LPS stage as they
are likely to develop at an early stage of fold growth (Tavani et al., 2015a).
*Folding* stage: set J2 reflects local extension associated with strata curvature at fold hinges. The
extensional trend, hence the trend of J2 joints/veins, changes as a function of curvature of fold axes
in map view. We also interpret the stylolite peaks of which orientation are intermediate between set
S1 and S2 (Fig. 4) as related to the folding stage (Roure et al., 2005).
*Late stage fold tightening (LSFT)*: some stylolites with peaks striking NE-SW (set S2) and some
veins/joints (set J3) postdate strata tilting and are consistent with late folding strike-slip and reverse
faults (set F2). We gathered these sets as markers of a late stage of fold tightening (LSFT), *i.e.* with
mesostructures still forming in response to contraction but slightly after fold growth ended.

It is noteworthy that the few occurrences of N-S striking veins/joints which are pre-folding and oblique
to the fold axis (Fig. 4) could be tentatively related to a still earlier stage of *foreland flexure and bulging*,
*i.e.* foredeep-parallel stretching associated with lithosphere flexuring (Tavani et al., 2013), even
though the lithospheric forebulge was described only far east of the central Apennines area (Tavani
et al., 2015b). However, these fractures, described in the Monte Nero (Beaudoin et al., 2016) and in



the Monte Catria (Tavani et al., 2008), were not interpreted by the authors because flexure/forebulge
has never been recognized in the UMAR. We will not discuss these joints further.

b.   Burial depth evolution and timing of contractional deformation

Stylolite roughness inversion applied to bedding-parallel stylolites (BPS) provides access to the
maximum depth experienced by the strata at the time vertical shortening was prevailing on horizontal
shortening, while $\sigma_1$ was vertical (Ebner et al., 2009b;Koehn et al., 2007;Beaudoin et al.,
2019;Beaudoin et al., 2016;Beaudoin and Lacombe, 2018;Beaudoin et al., 2020;Rolland et al.,
2014;Bertotti et al., 2017). In this study, we propose to compare the depth range returned by the
inversion of a population of BPS to a local burial model (Fig. 10) reconstructed from the strata
thickness documented in wells located in the western-central part of the UMAR (Nero-Catria area)
(Centamore et al., 1979;Tavani et al., 2008). The timing of exhumation was constrained by published
paleogeothermometric studies and by sedimentary records (Caricchi et al., 2014;Mazzoli et al., 2002).
To the West, tectonic reconstructions and organic matter paleothermometry applied to the Tuscan
Nappe (Caricchi et al., 2014) revealed that most of this unit underwent abnormal burial because it was
underthrusted below the Ligurian Nappe, but that the western front of the Ligurian Nappe did not
reach Monte Corona (Caricchi et al., 2014). We therefore consider a unique burial curve for the whole
UMAR, and we project the range of depth values at which individual BPS stopped being active on the
burial curves of the formations hosting the BPS. Recent application of this technique, coupled with
absolute dating of vein cements (Beaudoin et al., 2018), showed that the greatest depth that a
population of BPS recorded was reached nearly at the time corresponding to the age of the oldest LPS-
related veins, suggesting that it is possible to constrain the timing at which horizontal principal stress
overcame the vertical principal stress, switching from burial-related stress regime ($\sigma_1$ vertical) to LPS
($\sigma_1$ horizontal) (Beaudoin et al., 2020). In the case of the UMAR, this projection highlights that the BPS
population started to stop being active at a depth as shallow as 800m in all studied formations,
confirming that burial-related pressure solution (*i.e.*, chemical vertical compaction) initiated at even
shallower depths (Ebner et al., 2009b;Rolland et al., 2014;Beaudoin et al., 2019;Beaudoin et al., 2020).

Figure 10 also shows that BPS were active mainly from the Cretaceous (age of deposition of
the platform) until Serravallian times (~12 Ma), which suggests that LPS started around that time. As
the sedimentary record pins the beginning of folding of the UMAR to the Tortonian in the west and to
the Messinian in the east (onshore) (Calamita et al., 1994), we propose that, as an average, in the
central and western part of the UMAR, the LPS stage of Apennine contraction lasted about ~4 Ma
(Langhian to Tortonian) before folding occurred. Absolute dating of faults related to late stage fold
tightening in the central part of the UMAR further indicates that fold development was over by the
beginning of the Pliocene (~5 Ma). We can therefore estimate the duration of folding in the western-
central part of the UMAR to ~3 Ma. Knowing the oldest record of post-orogenic extensional tectonics
in the UMAR is mid-Pliocene (~3 Ma) (Barchi, 2010), we can also estimate the duration of the LSFT to
~2 Ma. In total, the period of time when the compressive horizontal principal stress $\sigma_1$ was higher in
magnitude that the vertical stress (*i.e.* until post-orogenic extension) lasted for 9 Ma in the Western-
Central part of the UMAR. This 3 Ma can be considered as the average duration of the individual fold
growth (~3 Ma), thus can be compared to the few attempts previously made to reconstruct the
duration of fold growth. Using syntectonic sedimentation, various studies reconstructed constant fold
growth lasting from between 3 to 10 Ma (Anastasio, 2007;Holl and Anastasio, 1993), up to 24 Ma with



quiescent periods in between growth pulses (Masaferro et al., 2002). From mechanical or kinematic
modeling applied to natural cases, reconstructed folding duration range from 1 Ma to 8 Ma (Suppe et
al., 1992;Yamato et al., 2011). The combination of bedding-parallel stylolite inversion, burial models
and U-Pb dating of vein cements/fault coatings yields a valuable insight into the timing of the different
stages of contraction in a fold-and-thrust belt (Beaudoin et al., 2018), quite in accordance with
previous attempts to constrain fold growth duration and rates.

c.  Paleofluid origin, precipitation temperature and pathways
i.  Fluid system evolution

The combined use of BPS inversion and burial curves therefore constrains the absolute timing
of LPS in the UMAR (Fig. 10). The further combination of the timing of LPS with the knowledge of the
past geothermal gradient as reconstructed from organic matter studies in the eastern part of the TN
(23°C/km, Caricchi et al., 2014) therefore yields the expected temperature within the various strata
during the opening of the vein sets J1, J2 and J3, and faults F1 and F2. This makes it possible to identify
potential fluids having precipitated in veins in thermal disequilibrium with the host rocks, e.g., of
hydrothermal nature, during the Apenninic contraction, for all studied veins. The reconstructed
temperatures of precipitation, and the maximum temperatures predicted by the burial model as well,
are in agreement with the fact that most twins are thin (thickness < 5 $\mu$m) and rectilinear, suggesting
deformation at temperature below 170°C (Ferrill et al., 2004;Lacombe, 2010). In spite of the Sr
radiogenic signatures of the veins, that all fall into the range of expected values in the host rocks (Fig.
7) (McArthur et al., 2001), hinting for very limited exchange between reservoirs, geochemical datasets
altogether discriminate two different fluid flow history at the belt scale: the folds at the TN-UMAR
transition, *i.e.* Monte Corona and Monte Subasio, clearly exhibit a singular history compared to the
other folds of the UMAR (Fig. 6d):

- in the UMAR, data suggest that during LPS, the fluid system mainly involved local fluids that
mildly interacted with host rocks ($\delta^{18}O_{fluids} \approx 5‰$ VSMOW) and precipitated between 30°C and 50°C
(Fig. 8), *i.e.*, at thermal equilibrium considering a depth of 1 to 1.7 km predicted at the time of LPS
(Figs. 6, 10), and considering a surface temperature of 10°C and a geothermal gradient of 23°C/km
(Caricchi et al., 2014). We interpret these fluids as local formational fluids (re)mobilized during
pressure solution, burial and tectonic compaction and fracturing. During folding, fluid precipitation
higher temperature (70°C) and higher degree of fluid rock interaction ($5 < \delta^{18}O_{fluids} < 10‰$ VSMOW),
are consistent with the expected temperature at the depth of burial of the Scaglia Fm. in the Tortonian
(*i.e.*, at the time of folding), suggesting again a local source of fluids without significant migration at
the reservoir scale (Fig. 6c-d). Previously published isotopic and thermometric data for contractional
fluids flow in the easternmost part of the UMAR reported infill of hydrothermal (100°C) dolomitizing
fluid flow during contraction (Mozafari et al., 2019;Storti et al., 2018). These hydrothermal
dolomitizing fluids have the same range of signatures of $\delta^{18}O_{fluids}$ than the ones precipitating at the
thermal equilibrium we document in the other folds of the UMAR (except monte Subasio). That
suggests that the fluid system is rather local, with potential, local but seldom influence of faults to
connect strata to deeper Jurassic reservoirs. During LSFT, the cement coating of faults F2 returns
precipitation temperature of ca. 40°C to 70°C (Fig. 8), a temperature in line with the expected depth
during the LSFT (Fig. 9).

- At the transition between the TN and the UMAR, Monte Subasio and Monte Corona both
exhibit a similar fluid system evolution. During LPS, $\Delta_{47}CO_2$ and $\delta^{18}O$ signatures of vein cements and





fault coating show a variability of temperature of precipitation and origin of fluids. Two different fluids
can be defined (Fig. 6): (1) fluids precipitating at 50°C-70°C, *i.e.* at thermal equilibrium with the host
rock, with $\delta^{18}O$ signatures of the fluids ranging from 0‰ to 5‰ SMOW, supporting a local formational
marine source with no to small-scale migration, that precipitated has cements characterized by $\delta^{18}O$
between 0‰ and -10‰ PDB; (2) fluids that precipitated at 110°C to 140°C, *i.e.* hydrothermal, with
very high $\delta^{18}O$ signatures of the fluids (up to 15‰ SMOW) that precipitated has cements characterized
by very depleted $\delta^{18}O$ signatures (down to -17‰ PDB), and witnessing a migration of basinal brines,
further supported by more radiogenic signatures of the $^{87}Sr/^{86}Sr$ ratios in Monte Subasio. During
folding, we also document the hydrothermal fluids, but also fluids characterized by negative $\delta^{18}O_{fluids}$
that precipitated at a temperature consistent with predicted depth, interpreted as an input of
meteoric water through fractures in the reservoir. Note that the hydrothermal dolomitizing fluids
documented at the Montagna dei Fiori (Mozafari et al., 2019;Storti et al., 2018); have an isotopic
signature much lower (6‰ SMOW) than the ones from the fluids involved in the Monte Subasio and
Monte Corona (15‰ SMOW), supporting that a different fluid system was prevailing in this part of the
belt during LPS, folding and LSFT.

ii.   Fluid origin and engine of migration at the transition between the Tuscan
Nappe and the Umbria-Marches Arcuate Ridge

During LPS and folding, the concomitant high temperatures of precipitation (>100°C) and the
very positive O isotopic signatures of fluids ($\delta^{18}O_{fluids}$ > 10‰ VSMOW) indicate that the system was
locally overprint either with formational-derived hydrothermal fluid migrating from depth > 4 km, or
with hydrothermal Triassic fluids that have a very depleted original $\delta^{18}O_{fluids}$. Because the $^{87}Sr/^{86}Sr$
isotopic ratio is affected by neither fluid-rock interactions nor temperature changes, the radiogenic
values of $^{87}Sr/^{86}Sr$ can help discriminate between both sources. In the present case, our data lead to
discard the case where the fluids originated from Lower Triassic rocks and were remobilized during
LPS. Indeed, expected $^{87}Sr/^{86}Sr$ values of lower Triassic seawater are significantly higher (0.7080-
0.7082, (McArthur et al., 2001) than the $^{87}Sr/^{86}Sr$ values recorded by the fluids precipitating in the
Monte Subasio (0.7076-0.07077) (Fig. 7). This range of radiogenic signature rather points out that the
fluids were either formational fluids originating from the Scaglia rossa, that directly overlies the host
rock, or local formational fluids that interacted with the clay fraction of the host-rocks. The
coexistence inside a single deformation stage (LPS or folding) of both local/meteoric fluids and
hydrothermal brines migrated from depths can be explained by transient flush into the system of
hydrothermal fluids flowing from deeply buried part of the same, stratigraphically continuous,
reservoir (Bachu, 1995;Garven, 1995;Machel and Cavell, 1999;Oliver, 1986).
We propose that the fluid system prevailing at the Monte Corona and at the Monte Subasio
reflects an eastward, squeegee-type, flow of hydrothermal fluids (Fig. 11), for which the long-term
migration engine is the lateral variation of the depth of the reservoir, buried under the stacked Tuscan
and Ligurian Nappes in the west (up to 4 km, Caricchi et al., 2014), while just buried under the
stratigraphic succession in the east (up to 2.5 km, Fig. 11b). This depth variation likely created a water
table top difference in height, and so an hydraulic gradient allowing for the eastward fluid migration
within the reservoir, enhanced by LPS and related fracture development (Roure et al., 2005). As the
paleodepth variation was related to the weight of the nappes stacking rather than to a foreland-type
slope, the UMAR would then have formed a plateau without any large-scale lateral fluid migrations



(Fig. 11b). The inferred pulses of hydrothermal fluids also implies a strong influence of forelandward
propagation of contractional deformation in the eastward fluid expellation (Oliver, 1986;Machel and
Cavell, 1999).

603         d.   Influence of tectonic style on fluid flow during deformation history

Our study of the calcite cements that precipitated in tectonically controlled veins and faults at the
scale of the UMAR and TN distinguishes two different fluid flow histories. East of Monte Subasio, *i.e.*,
in the UMAR where shortening is distributed on deep-rooted faults, our data reveal a closed fluid
system, with formational fluids precipitating at thermal equilibrium, limited fluid - host rock
interactions in the reservoirs and limited cross-stratal fluid migration. In contrast, on the western part
of this divide (in the TN), where shortening was accommodated by stacked nappes detached above
the Triassic décollement level, high temperatures of fluids suggest the occurrence of eastward large-
scale pulses of hydrothermal fluids (squeegee type, Fig. 11b).
If considering a thin-skinned tectonic model for the UMAR with shallow, low angle thrusts rooting
on the Triassic evaporitic décollement level (Fig. 1) (Bally et al., 1986), one would expect some
signature of Triassic fluids to be involved in the reservoir paleohydrology at the time faults were active
or during folding, as illustrated in similar salt-detached fold systems in the Pyrenees, in the
Appalachians, and in the Sierra Madre Oriental  ((Lacroix et al., 2011;Travé et al., 2000;Evans and
Hobbs, 2003;Evans and Fischer, 2012;Fischer et al., 2009;Smith et al., 2012;Lefticariu et al., 2005). One
the other hand, if considering a thick-skinned tectonic model with high angle thrusts crossing the
Triassic down to the basement, it becomes more likely that these thrusts did not act as efficient
conduits for deep fluids (evaporitic fluids or basement fluids) as fault damage zones in evaporites
remains non permeable, and if the displacement along the faults is smaller than the nonpermeable
layer thickness. This contrasts with paleohydrological studies of basement cored folds, where high
angle thrusts allow hot flashes of hydrothermal fluids into the overlying cover (Beaudoin et al.,
2011;Evans and Fischer, 2012) in the absence of evaporites. Thus the lack of Triassic signature in our
paleofluid dataset seems to support a thick-skinned tectonic style of deformation in the UMAR Fig.
11c).This fluid flow model therefore outlines important differences between belts where shortening
is localized and accommodated by nappe stacking, typical from thin-skinned belts, and belts where
shortening is instead distributed on several folds related to high angle thrusts, typical of thick-skinned
belts (Lacombe and Bellahsen, 2016). Squeegee-type fluid flow during LPS in response to hydraulic
gradient and lateral tectonic contraction has also been described in other thin-skinned belts, such as
the Canadian Rocky Mountains (Vandeginste et al., 2012;Roure et al., 2010;Machel and Cavell,
1999;Qing and Mountjoy, 1992), or in Venezuela (Schneider et al., 2002;Schneider et al., 2004;Roure
et al., 2003) where lithospheric bulging was the origin of the depth difference leading to hydraulic
gradient-driven migrations. The presented case study shows how stacking of sedimentary units typical
of thin-skinned tectonics strongly influences the fluid system beyond the morphological front of the
belt, and can allow large scale fluid migrations even in the absence of (well-expressed) lithospheric
forebulge occurred.
4.   Conclusions
Our study of the vein-fault-tectonic stylolite populations distributed in Jurassic to Eocene
limestone rocks at the scale of the thin-skinned Tuscan Nappe and presumably thick-skinned Umbria-



Marche Apenninic Ridge reveals the occurrence of several fracture/stylolite sets that support a three stages evolution of the Apenninic contraction : (1) layer-parallel shortening is reconstructed by a set of joint/veins striking NE-SW to E-W, perpendicular to the local trend of the fold, alongside with stylolite peaks striking NE-SW, and early folding bedding-parallel reverse faults; (2) folding stage is recorded by fold-parallel mode I joints and veins; (3) late stage fold tightening is recorded by late post-tilting, late folding stylolite peaks, joints/veins and also mesoscale reverse and strike-slip faults.

Thanks to burial models coupled to bedding-parallel stylolite paleopiezometry, along with (unfortunately scarce) U-Pb absolute dating of strike-slip faults related to late stage fold tightening, we were able to reconstruct the timing of the onset and the duration of the Apennine contraction, with an unparalleled detail: the LPS started by Langhian time (~12 Ma, inferred from the bedding parallel stylolite inversion), lasted for ~4 Ma, then folding started by the Tortonian time (8 Ma, from published syn-tectonic sedimentary constraints), lasted for ~3 Ma, LFST started by the beginning of Pliocene (5 Ma, given by absolute dating of fault coatings), itself lasting for 2 Ma before post-orogenic extension affected strata since mid-Pliocene (3 Ma).

Accessing the starting and ending timing of deformation in the UMAR also allowed us to predict the depth and expected temperatures of the paleofluid during fracturing assuming fluids precipitated at thermal equilibrium. By characterizing the cements related to sets of veins and faults using O and C stable isotope signatures, radiogenic signatures of $^{87/86}$Sr, and clumped isotopes of $\Delta^{47}CO_2$, we show that different paleofluid systems occurred during LPS and folding from west to east of the section. In the westernmost folds of the UMAR located beyond the arrow of the Ligurian Nappe thrusting over the Tuscan Nappe, we highlighted a local fluid system with transient flush of large-scale lateral, stratigraphically-compartimentalized migration of hydrothermal fluids. In contrast, these pulses are not documented in the rest of the UMAR and its foreland, where the fluid system remained closed at all time. We tentatively relate this change in fluid system to a lateral change in tectonic style of deformation across the belt, from thin-skinned in the TN to rather thick-skinned in the UMAR. Beyond regional implications, this study highlights the potential of such multi-proxy approach to unravel coupled structural and fluid flow evolution in fold-and-thrust belts.

Author contribution

NB, OL, DK, A. Billi, JPC were involved in the overall writing of the manuscript led by NB; NB, OL, DK, A. Billi collected structural data and rock samples in the field; NB, AL and OL conducted microstructural inversion; GH, A. Boyce, CJ, MM, NR, IM, FC and CP designed experiments and collected the geochemical data and wrote the related parts of the manuscript and appendices. All authors critically reviewed the multiple drafts of the manuscript.

5. Acknowledgments

This work has received funding from the Natural Environmental Research Council under grant number IP-1494-1114, from European Union's Seventh Framework Program for research, technological development, and demonstration under grant agreement 316889, and it also received funds from Sorbonne Université (research agreement C14313). NB is funded through the ISITE program E2S, supported by ANR PIA and Région Nouvelle-Aquitaine.

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

Figures with captions

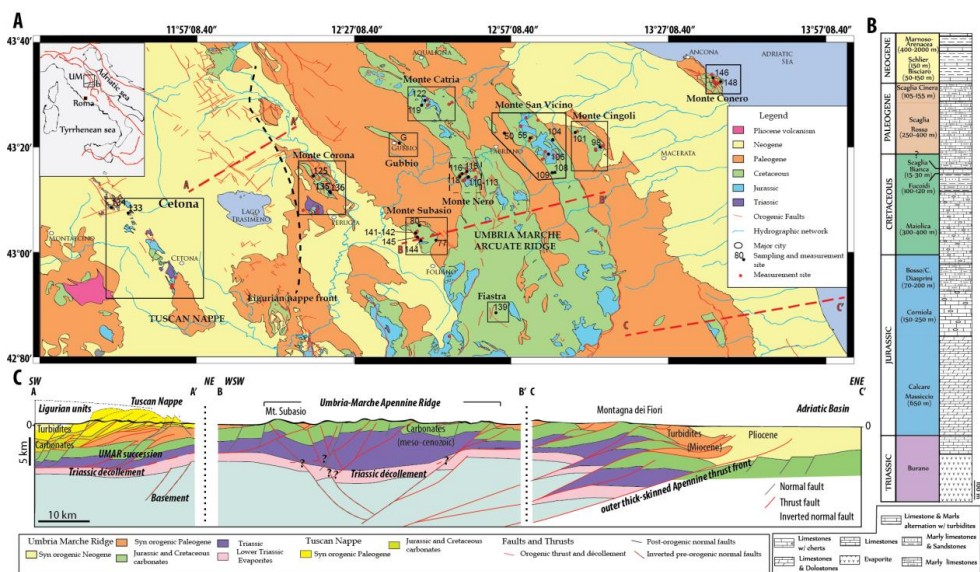


Figure 1: Simplified geological map of the study area, with located the sampling and measurement
sites. Frames relate to the fracture study areas used in figure 4. Exact location of measurement sites
are reported as black and red points, and labelled black points also represent the sampling site for
geochemical analysis. B. Stratigraphic column based on stratigraphic and well data from the central





part of the UMAR, after Centamore et al. (1979). C. Crustal-scale composite cross-section based on
published seismic data interpretations, A-A' modified after Carboni et al. (2020); B-B' and C-C' after
Scisciani et al. (2014). Note that both tectonic style (thick-skinned and thin-skinned) are represented
by question marks for the UMAR.

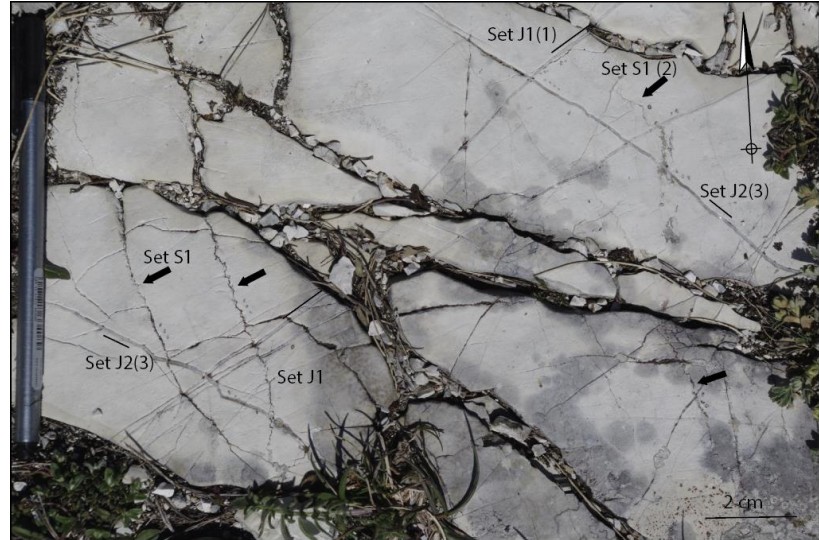

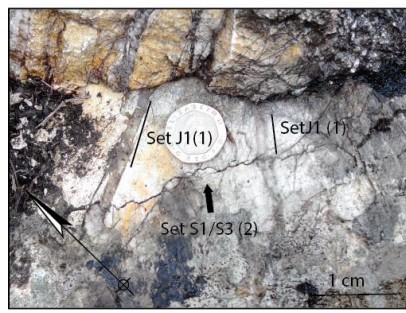
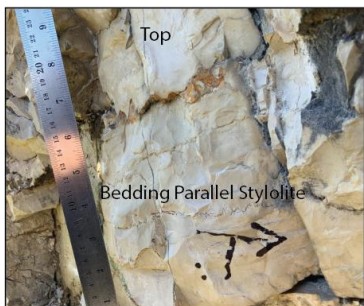


Figure 2: Field photographs showing chronological relationships between veins/joints and stylolites.
A) Monte Nero, b) Monte Cingoli, c) Monte Subasio. Sets are reported along with local chronological
order between brackets.



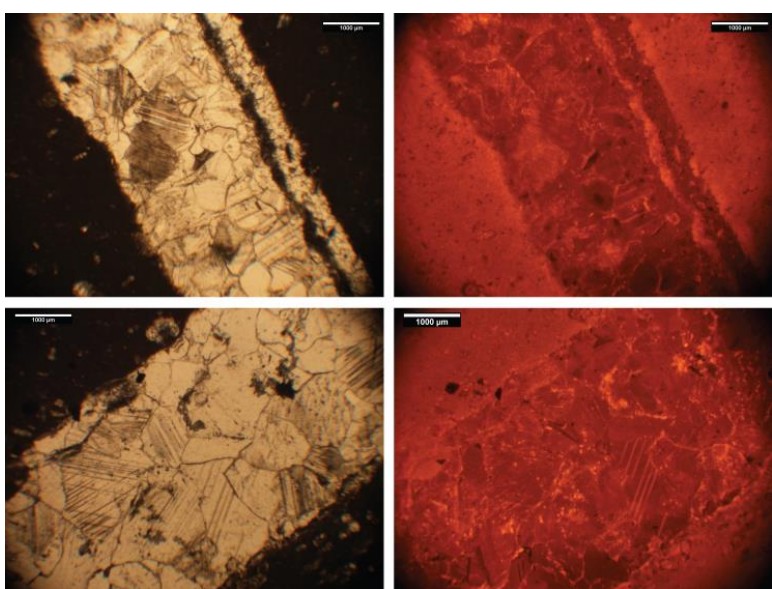


Figure 3: photomicrographs of various veins in natural light (left-hand side), with corresponding view
under cathodoluminescence (right-hand side), top one is a set J1 vein from the Scaglia Fm., bottom
one is a set J2 from the Maiolica Fm.

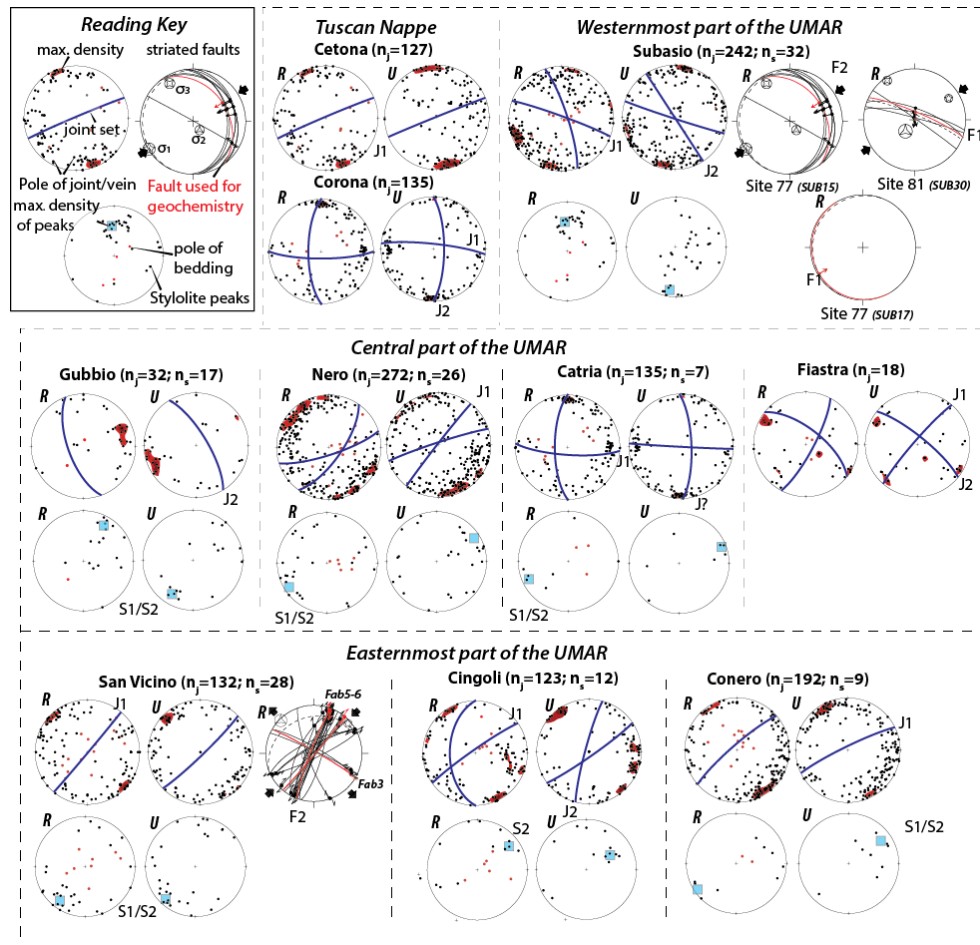

Figure 4: Poles of measured joints/veins and stylolite peaks projected on Schmidt stereograms, lower hemisphere, for the different structures. Data are projected in the current attitude of the strata (left, R), with pole to bedding in red, and after unfolding (right, U). Red color scale represents highest density according to Fischer statistical analysis using the software OpenStereo, and main fracture set average orientation are represented as blue planes. For tectonic stylolite peaks, the blue square represents highest density according to Fischer statistical analysis. Striated fault inversion results are reported in the current attitude of the strata (bedding as dashed line).

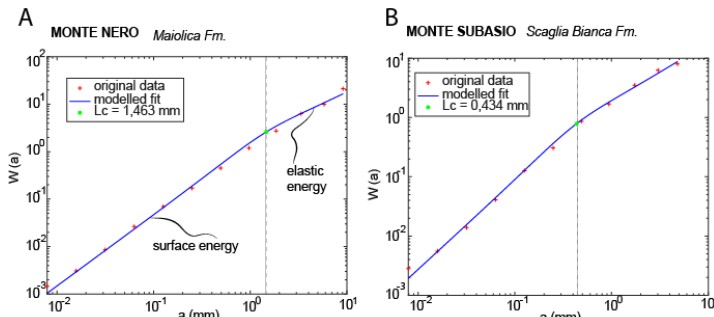


Figure 5: Examples of results of stylolite roughness inversion, with signal analysis by Average Wavelet
in the Monte Nero (A) and in the Monte Subasio (B).

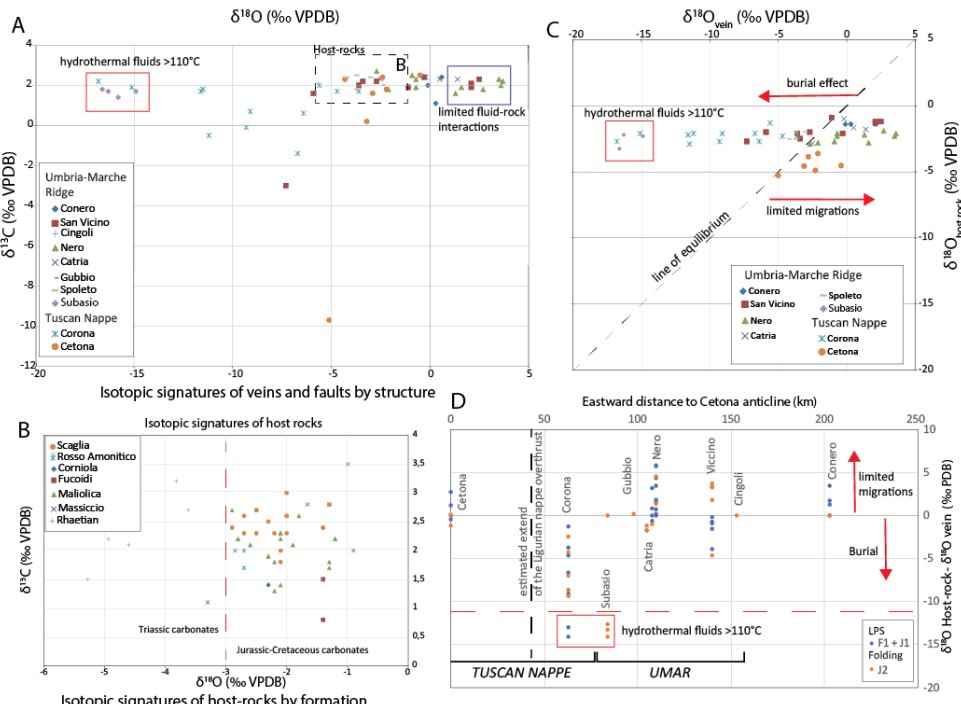


Figure 6: A) Plot of δ¹³C vs δ¹⁸O (‰ VPDB) of veins represented by structure. Frames represent the
different type of fluid system. B) Plot of δ¹³C vs δ¹⁸O (‰ VPDB) of host-rocks represented by structure.
C) Plot of δ¹⁸Ovein vs δ¹⁸Ohost (‰ VPDB) of veins represented by structures. D) Plot of the difference
between δ¹⁸Ohost-rocks and δ¹⁸O veins (‰ VPDB) vs eastward distance from the Cetona Antincline
towards the Adriatic basin across the strike of the UMAR. Data are represented by tectonic sets. The
proposed extension of the Ligurian nappe overthrust is reported after Caricchi et al., 2014; red frames,
arrows and lines represent the fluid systems.



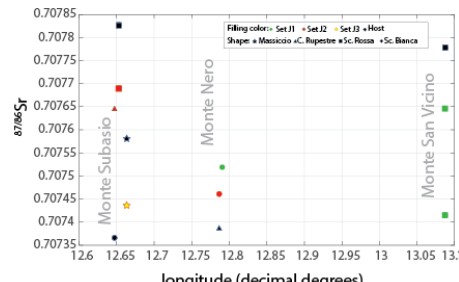

Figure 7: Plot of $^{87/86}$Sr signatures vs longitude, with filling color related to tectonic set and point shape related to host rock formation.

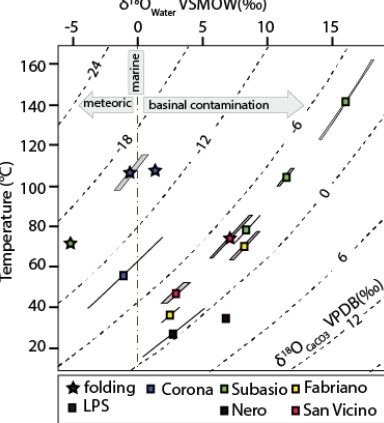

Figure 8: Plot of $\delta^{18}O_{fluid}$ (‰ SMOW) vs precipitation temperature (°C) obtained from clumped isotope analyses, oblique lines are the measured $\delta^{18}O_{CaCO3}$ of the vein cements (‰ PDB). Shape of the points correspond to tectonic set (LPS being U1 and compatible faults and folding U2), while filling color relates to structure. Oblique dotted lines are the measured $\delta^{18}O$ signatures of carbonates.

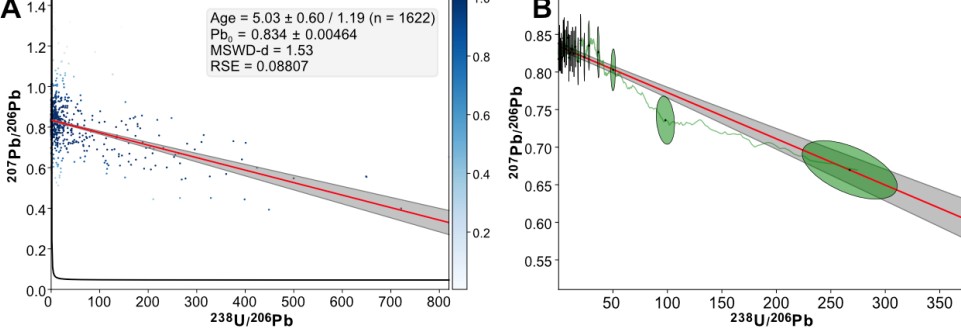

Fig. 9: A) Tera-Wasserburg concordia plot obtained from LA-ICPMS U-Pb dating of FAB5 calcite sample. The age was obtained by robust regression through the U-Pb image pixel values. The scale bar corresponds to the weight of each pixel as determined by robust regression. B) Same plot but with





discretized data represented as ellipses (one ellipse = 60 pixels). The running mean (window = 60
pixels) is also shown as green line.

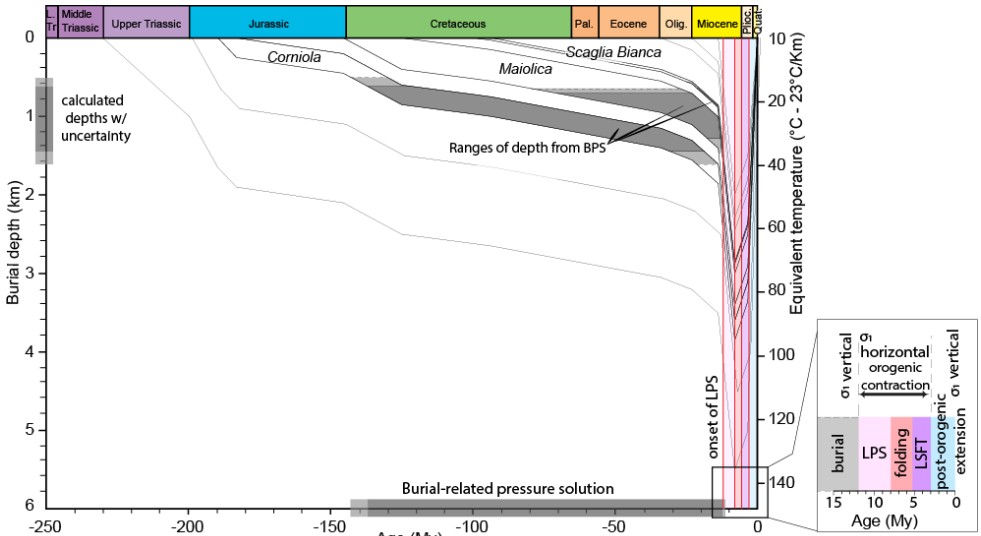

Figure 10: Burial model valid for the Umbria Marche Ridge, derived from well data and previously
published burial models in the Tuscan Nappe (Caricchi et al., 2015). The range of depths reconstructed
from bedding parallel stylolite roughness inversion (with uncertainty) are reported for each formation
as grey shades. The derived corresponding timing and depth of active dissolution are reported on the
x-axis and left y-axis, respectively. The timing of the deformation is reported on the right-hand side as
a zoom. The onset of Layer Parallel shortening is deduced from the latest bedding stylolite to have
been active, the onset of LSFT is given by U-Pb dating of fault coating in this study. The timing of folding
and post-orogenic extension are reported from published sedimentary data (see text for more
detailed explanations).





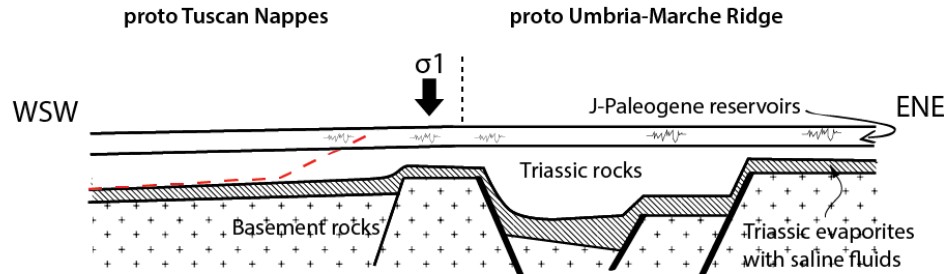

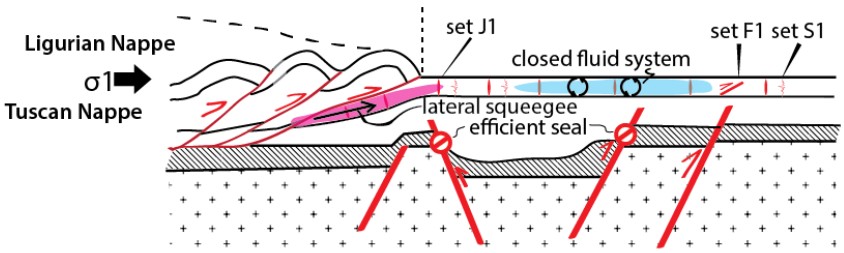

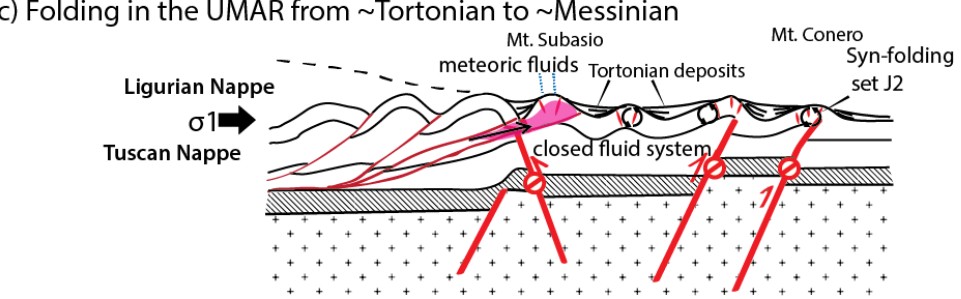


Figure 11: Conceptual model representing fracture development and regional scale fluid dynamics
during the formation of the Tuscan Nappes and Umbria Marche Ridge. Red areas represent the extent
of pulses of eastward hydrothermal fluids. Blue areas represent closed fluid system at the scale of the
carbonate reservoirs.











Table 1 - Results of Stylolite Roughness Inversion applied on beddIng-parallel stylolites

| Sample | GPS | location | formation | Lc (mm)* | E (GPa) | mu | v | vertstress (Pa) | depth (m) |
|--------|-----|----------|-----------|----------|---------|------|------|-----------------|-----------|
|        | 165 | Subasio | Scaglia Bianca | 1,059 | 23,2 | 0,25 | 0,32 | 21811000 | 926 |
|        | 165 | Subasio | Scaglia Bianca | 1,306 | 23,2 | 0,25 | 0,32 | 19640000 | 834 |
|        | 165 | Subasio | Scaglia Bianca | 0,46 | 23,2 | 0,25 | 0,32 | 33093000 | 1406 |
| A165   | 165 | Subasio | Scaglia Bianca | 0,486 | 23,2 | 0,25 | 0,32 | 32196000 | 1368 |
|        | 165 | Subasio | Scaglia Bianca | 0,434 | 23,2 | 0,25 | 0,32 | 34070000 | 1447 |
|        | 165 | Subasio | Scaglia Bianca | 0,971 | 23,2 | 0,25 | 0,32 | 22778000 | 967 |
|        | 165 | Subasio | Scaglia Bianca | 1,488 | 23,2 | 0,25 | 0,32 | 18400000 | 782 |
|        | 110 | Nero | Maiolica | 1,073 | 23,2 | 0,25 | 0,32 | 21668000 | 920 |
|        | 110 | Nero | Maiolica | 1,535 | 23,2 | 0,25 | 0,32 | 18116000 | 769 |
|        | 110 | Nero | Maiolica | 1,463 | 23,2 | 0,25 | 0,32 | 18557000 | 788 |
|        | 110 | Nero | Maiolica | 1,071 | 23,2 | 0,25 | 0,32 | 21688000 | 921 |
| AN26   | 110 | Nero | Maiolica | 1,29 | 23,2 | 0,25 | 0,32 | 19762000 | 839 |
|        | 110 | Nero | Maiolica | 1,073 | 23,2 | 0,25 | 0,32 | 22661000 | 962 |
|        | 110 | Nero | Maiolica | 1,596 | 23,2 | 0,25 | 0,32 | 17767000 | 755 |
|        | 110 | Nero | Maiolica | 0,659 | 23,2 | 0,25 | 0,32 | 27649000 | 1174 |
|        | 110 | Nero | Maiolica | 0,696 | 23,2 | 0,25 | 0,32 | 26904000 | 1143 |
| AN16   | 115 | Nero | Maiolica | 1,279 | 23,2 | 0,25 | 0,32 | 19847000 | 843 |
| A137   | 148 | Conero | Scaglia Bianca | 2,073 | 23,2 | 0,25 | 0.32 | 15589000 | 662 |
| A123-2 | 130 | Gubbio | Corniola | 0,428 | 23,2 | 0,25 | 0,32 | 34308000 | 1457 |
|        | 130 | Gubbio | Corniola | 0,791 | 23,2 | 0,25 | 0,32 | 25237000 | 1072 |
| A123   | 130 | Gubbio | Corniola | 2,35 | 23,2 | 0,25 | 0,32 | 14642000 | 622 |
|        | 130 | Gubbio | Corniola | 1,457 | 23,2 | 0,25 | 0,32 | 18595000 | 790 |
| A21    | 104 | San Vincino | Maiolica | 0,906 | 23,2 | 0,25 | 0,32 | 23581000 | 1002 |
|        | 104 | San Vincino | Maiolica | 0,787 | 23,2 | 0,25 | 0,32 | 25414000 | 1079 |
|        | 138 | Spoleto | Scaglia Bianca | 0,655 | 23,2 | 0,25 | 0,32 | 27733000 | 1178 |
|        | 138 | Spoleto | Scaglia Bianca | 0,634 | 23,2 | 0,25 | 0,32 | 28189000 | 1197 |
| A104   | 138 | Spoleto | Scaglia Bianca | 0,66 | 23,2 | 0,25 | 0,32 | 27628000 | 1174 |
|        | 138 | Spoleto | Scaglia Bianca | 1,22 | 23,2 | 0,25 | 0,32 | 20321000 | 863 |
|        | 138 | Spoleto | Scaglia Bianca | 0,749 | 23,2 | 0,25 | 0,32 | 25935000 | 1102 |
|        | 138 | Spoleto | Scaglia Bianca | 1,322 | 23,2 | 0,25 | 0,32 | 19521000 | 829 |

* the crossover length is given within 12% uncertainy, using values for Young Modulus (E) of 23,2 Gpa
(Beaudoin et al., 2014), Poisson ratio (mu) of 0,25 and an interfacial energy (v) of 0,32 J.m$^{-2}$











Table 2 - Results of stable isotopic signature of O, C, and radiogenic signatures of Strontium $^{86}Sr/^{87}Sr$

| Sample | GPS | Formation | Structure | Set | $\delta^{18}O$ Vein ‰V-PDB | $\delta^{13}C$ Vein ‰V-PDB | $\delta^{18}O$ HR ‰V-PDB | $\delta^{13}C$ HR ‰ V-PDB | $^{87}Sr/^{86}Sr_V$ | $^{87}Sr/^{86}Sr_{Hr}$ |
|---|---|---|---|---|---|---|---|---|---|---|
| A94V | 134 | Retian * | Cetona | J1 | -3,2 | 0,2 | -3,2 | -4,6 | | |
| A93V | 134 | Retian * | Cetona | J1 | -2,9 | 1,6 | -2,5 | -3,82 | | |
| A95V | 134 | Retian * | Cetona | J1 | -0,5 | 2,5 | -3,2 | -4,53 | | |
| A92V | 134 | Retian * | Cetona | J1 | -4,3 | 2,3 | | | | |
| A89V | 133 | Retian * | Cetona | J1 | -2,4 | 2,4 | -3,6 | -4,93 | | |
| A84V | 133 | Retian * | Cetona | J2 | -2,2 | 1,8 | -2,3 | -3,62 | | |
| A86V | 133 | Retian * | Cetona | J2 | -5,1 | -9,7 | -3,9 | -5,28 | | |
| A76F | 125 | Maliolica | Corona | F1 | -6,7 | -1,4 | -2,1 | 2,3 | | |
| A76V2 | 125 | Maliolica | Corona | J1 | -15,1 | 1,9 | -2,1 | 2,3 | | |
| A76V1 | 125 | Maliolica | Corona | J1 | -11,2 | -0,5 | -2,1 | 2,3 | | |
| A72V | 125 | Maliolica | Corona | J2 | -5,6 | 2 | -1,3 | 1,8 | | |
| A76V3 | 125 | Maliolica | Corona | J2 | -9,1 | 0,7 | -2,1 | 2,3 | | |
| A77V2 | 125 | Maliolica | Corona | J2 | -11,6 | 1,7 | -2,2 | 1,8 | | |
| A77V1 | 125 | Maliolica | Corona | J2 | -11,5 | 1,8 | -2,9 | 2,7 | | |
| A96V | 135 | Rosso Amonitico | Corona | J1 | -16,8 | 2,2 | -2,7 | 2 | | |
| A97bV1 | 135 | Rosso Amonitico | Corona | J1 | -6,4 | 0,6 | -2,7 | 1,7 | | |
| A97bV2 | 135 | Rosso Amonitico | Corona | J1 | -9,3 | -0,1 | -2,7 | 1,7 | | |
| A98V1 | 136 | Corniola | Corona | J1 | -3,6 | 1,7 | -2,3 | 1,4 | | |
| A98V2 | 136 | Corniola | Corona | J2 | -4,7 | 1,7 | -2,3 | 1,4 | | |
| A121V | G | Maliolica | Gubbio | J2 | -2,4 | 2 | -2,6 | 2,1 | | |
| A112V1 | 141 | Massiccio | Subasio | J2 | -15,8 | 1,4 | | | | |
| A111V | 141 | Massiccio | Subasio | J2 | -16,6 | 1,8 | -3,3 | 1,1 | 0,707644 | 0,707366 |
| A118V | 145 | Scaglio Rossa | Subasio | J2 | -16,3 | 1,7 | -2,2 | 2,3 | 0,707690 | 0,707827 |
| A120V | 145 | Scaglia Rossa | Subasio | J2 | -14,9 | 1,7 | -2,3 | 2,5 | | |
| A116V | 144 | Massiccio | Subasio | J3 | | | | | 0,707437 | 0,707580 |
| A59V | 119 | Maliolica | Catria | J1 | 1,4 | 2,3 | -1,8 | 2,6 | | |
| A73V | 122 | Massiccio* | Catria | J1 | -0,2 | 2,4 | 0,4 | -0,99 | | |
| A63V | 122 | Massiccio* | Catria | J1 | 0,5 | 2,3 | -0,3 | -1,66 | | |
| A66V | 122 | Massiccio* | Catria | J2 | -2,5 | 2,3 | -1,5 | -2,85 | | |
| A56V | 118 | Scaglia Cinera | Nero | J1 | -0,9 | 2,5 | -2,7 | 2,3 | | |
| A57bV | 118 | Scaglia Cinera | Nero | J1 | -2,7 | 2,7 | -2,9 | 2,4 | 0,707461 | 0,707382 |
| A53V1 | 116 | Maliolica | Nero | J1 | -2,1 | 1,8 | -2,8 | 2,2 | 0,707519 | |
| A53V2 | 116 | Maliolica | Nero | J1 | 1,6 | 1,9 | -2,8 | 2,2 | | |
| A52V | 115 | Maliolica | Nero | J1 | 1,3 | 1,6 | -2,2 | 1,3 | | |
| A50V1 | 113 | Maliolica | Nero | J1 | -0,7 | 2,3 | -2,3 | 1,9 | | |
| A50V2 | 113 | Maliolica | Nero | J1 | 3,5 | 2,3 | -2,3 | 1,9 | | |
| A47V | 112 | Maliolica | Nero | J1 | 3,7 | 2,2 | | | | |
| A46V | 112 | Maliolica | Nero | J2 | 2,7 | 1,8 | -1,9 | 2,1 | | |
| A44V | 111 | Maliolica | Nero | J1 | 3,6 | 2 | -2,1 | 2,2 | | |





| | | | | | | | | | | |
|---|---|---|---|---|---|---|---|---|---|---|
| A43V | 110 | Maliolica | Nero | J2 | -0,7 | 1,9 | -2,1 | 1,4 | | |
| A107F | 139 | Scaglia Rossa | Spoleto | F1 | -4,2 | 2,4 | -2,5 | 2,7 | | |
| A107V | 139 | Scaglia Rossa | Spoleto | J1 | -3,7 | 2,5 | -2,5 | 2,7 | | |
| A104V1 | 139 | Scaglia Rossa | Spoleto | J2 | -3,7 | 1,8 | -2 | 2,6 | | |
| A27V | 106 | Massiccio | San Vicinno | J1 | -1,1 | 1,9 | -0,9 | 2 | | |
| A40F | 109 | Scaglia Bianca | San Vicinno | F1 | -5,9 | 1,6 | -2 | 3 | | |
| A38V | 109 | Scaglia Bianca | San Vicinno | J2 | -7,3 | -3 | -2,7 | 2,6 | 0,707646 | 0,707778 |
| A18V | 104 | Maliolica | San Vicinno | J1 | 2,1 | 1,9 | -1,3 | 1,7 | | |
| A74V1 | 104 | Maliolica | San Vicinno | J2 | 2,1 | 2,1 | -1,2 | 2,2 | | |
| A74V2 | 104 | Maliolica | San Vicinno | J2 | 2,5 | 2,3 | -1,2 | 2,2 | | |
| A32V | 108 | Scaglia Bianca | San Vicinno | J1 | -3,4 | 2,2 | -2,5 | 2,3 | 0,707415 | 0,707778 |
| A29V | 108 | Scaglia Bianca | San Vicinno | J1 | -3,6 | 2 | -2,1 | 1,8 | | |
| A34V | 108 | Scaglia Bianca | San Vicinno | J1 | -2,7 | 2,2 | -2 | 2,3 | | |
| A30V | 108 | Scaglia Bianca | San Vicinno | J2 | -0,3 | 2,4 | -2,1 | 2 | | |
| A14V | 101 | Scaglia | Cingoli | J2 | | | -1,3 | 2,8 | | |
| A129bF | 146 | Scaglia | Conero | F1 | 2,1 | 2 | -1,4 | 2,4 | | |
| A129bV | 146 | Scaglia | Conero | F1 | -0,1 | 2 | -1,4 | 2,4 | | |
| A126V | 146 | Scaglia | Conero | J2 | 0,6 | 2,4 | | | | |
| A133V | 148 | Fucoidi | Conero | J1 | | | -1,4 | 1,5 | | |
| A135V | 148 | Fucoidi | Conero | J1 | 0,3 | 1,1 | -1,4 | 0,8 | | |

*: Values were corrected to reflect the fact that host rocks is dolomite; HR stands for Host Rock, V stands for Vein















Table 3 - Fluid precipitation temperature of oxygen isotopic signature derived from clumped isotope analysis

| Sample Name | Structure | Set | Host formation | Mineralogy | Temperature (˚C) | MinT (˚C) | MaxT (˚C) | Fluid d18O VSMOW (mean) | Fluid d18O VSMOW (min) | Fluid d18O VSMOW (max) |
|---|---|---|---|---|---|---|---|---|---|---|
| A74A | Corona | J2 | Maiolica | Calcite | 106,4 | 98,0 | 115,4 | -0,5 | -1,8 | 0,9 |
| A77-130 | Corona | J2 | Maiolica | Calcite | 107,6 | 106,0 | 109,3 | 1,4 | 1,2 | 1,7 |
| A77-40 | Corona | J1 | Maiolica | Calcite | 55,9 | 39,9 | 74,7 | -1,1 | -3,9 | 1,9 |
| A120 | Subasio | J2 | Scaglia Rossa | Calcite | 71,7 | 71,7 | 71,7 | -5,2 | -5,2 | -5,1 |
| A28 | Subasio | J1 | Massiccio | Calcite | 119,1 | 114,8 | 123,6 | 11,3 | 10,7 | 12,0 |
| SUB15 | Subasio | F1 | Scaglia Cinerea | Calcite | 78,3 | 71,7 | 85,3 | 8,4 | 7,4 | 9,5 |
| SUB17 | Subasio | F1 | Scaglia Cinerea | Calcite | 140,9 | 122,9 | 161,8 | 16,1 | 14,0 | 18,2 |
| SUB30 | Subasio | F1 | Scaglia Cinerea | Calcite | 104,4 | 100,1 | 108,9 | 11,4 | 10,8 | 12,1 |
| A52 | Nero | J1 | Maiolica | Calcite | 34,7 | 33,9 | 35,5 | 6,8 | 6,6 | 7,0 |
| A56 | Nero | J1 | Scaglia Rossa | Calcite | 27,2 | 15,9 | 39,9 | 2,7 | 0,3 | 5,2 |
| A29 | San Viccino | J1 | Scaglia Rossa | Calcite | 47,3 | 42,5 | 52,4 | 3,0 | 1,9 | 4,0 |
| A39 | San Viccino | J2 | Scaglia Rossa | Calcite | 74,5 | 64,7 | 85,2 | 7,2 | 5,6 | 8,8 |
| FAB3 | San Viccino | F2 | Langhian Flysch | Calcite | 36,5 | 32,8 | 40,3 | 2,5 | 1,7 | 3,3 |
| FAB6 | San Viccino | F2 | Langhian Flysch | Calcite | 70,3 | 63,7 | 77,4 | 8,3 | 7,1 | 9,4 |

