# Peer review of "Regional-scale paleofluid system across the Tuscan Nappe Umbria Marche Arcuate Ridge (northern"

_Solid Earth, 2020_

## Referee Comment (RC1) · Anonymous Referee #1 · 7 Jun 2020

The authors present an interpretation of the paleofluid flow history across the Tucscan Nappe, constrained by a multiproxy study. They integrate detailed structural analysis of stylolites fracture networks with isotopic (O and C stable isotope, Sr isotopes, clumped carbonate isotopes) and some limited U-Pb dating of calcite veins, cements, and fault coatings. Using the dates from this study in concert with other timing and burial constrains, the authors reconstruct a $\sim$12 my history of fluid flow events that are linked to deformation. They identify evidence for both closed fluid systems and the open system flow of hydrothermal fluids.

This is an interesting and comprehensive evaluation of the role and evolution of fluids during regional tectonic compression. The interpretations appear mostly supported by the data, the data is of good quality, and the methods thorough in most cases. The data are high quality and this is certainly appropriate for publication in Solid Earth, and will be of interest to structural geologists, geochemists, and those working on problems in regional tectonics. One of the big contributions is the careful linkage between deformation and closed versus open-system fluid-rock interaction.

However, the reviewed version requires moderate revisions to improve the impact of the paper. I outline the major issues here, and then provide some line-by-line comments below.

1) Writing: This paper needs a thorough technical edit paying particular attention to grammar, tense, subject-verb agreement, over-reliance on the passive voice, general clarity of sentences. Also, it still reads like it was prepared by multiple authors with different styles, and without careful review. In the line-by-line comments I point out many of the grammatical problems, but by no means are these comprehensive.

2) Organization: The presentation style of method-results, method -results, etc. is quite distracting to this reviewer. The authors state in the paper that this is intentional with the goal of improving clarity. However, in order to follow all of the different and related data sets, it is very distracting to switch gears to reading another methods summary. True, I could skip over, but I would rather have the option of reading a single methods section, and a single results section. Also, this has resulted in some of the methods sections actually containing results, so they are merging together in some places. This is a relatively complicated data set to understand as a reader, so the organization needs to be strong.

3) Over-reliance on "isotopic jargon": For clarity to non-experts and experts alike, it is important to use proper language when presenting and interpreting the isotopic results. For example, the stable isotope community prefers writing "$\delta18O$ values". Avoid use

of depleted, enriched, heavy, and light…be specific. If using enriched or depleted, please be specific as to what is enriched with respect to what. Similarly, for the Sr - isotopes…..avoid the "radiogenic" term unless being specific – for example, rather than writing that a sample is less radiogenic than another, just provide the values.. Also, the term "signature" is really overused in this paper. Be careful with its use. A stable isotope delta value is not a signature…avoid "$\delta$18O signature of X ‰'. It is OK to use phrases like "meteoric isotopic signature".

4) Adding numeric ages to Geologic Ages: For those of us who do not work in the Miocene, referring to the Late Aquitanian and Messinian time lacks clarity. Please include the numeric ages as well. One, this helps the reader piece together the geologic history of your study, and it teaches some of use more details of the time scale!

5) Lack of adequate support/discussion for interpretations: The interpretation of the source fluids and degree of water-rock interaction are just stated. There is little discussion or justification and no examples from the literature. Please justify your interpretations. Also, the carbon isotope data is not discussed.

6) U-Pb dating – this is outside of my expertise, but my read is that these veins only yielded limited information?

7) Clumped isotope data: The $\Delta$47 results need to be added to Table 2 or 3, and presented in the results.

line by line

40-48: Paragraph needs strengthening…currently a 2 sentence paragraph with a really long second sentence. Suggest breaking this down for more impact.

53: comma before and

87: Define "LPS" here

90 – 91: mixing tense: are and was in same sentence

100: This reviewer cautions the claim of "provides for the first time"…this is rarely true.

120-121: please provide numeric dates along with "Late Aquitanian"…etc. Also, should Late be capitalized?

128: consider different and more active voice for "has been considered for long"

144: Replace "Nowadays" with Currently; "undergoes" with "is experiencing"

168 – 170: This reviewer finds this format difficult to follow.

179-180: This is actually a result, not a method.

186: need parentheses around sigma 1

197: How did you do the correction due to bedding dip – that is what program, or by hand…

212: Can be gathered? consider different word choice

221: New paragraph at "Finally"

255: Perhaps a complicated equation like this should be a numbered one.

288: Oxygen and Carbon should not be capitalized

293 – 303: need a method citation; specify dual inlet or continuous flow; probably not 105% or would crystallize…or did you measure density? How long were samples equilibrated at 90C?

306: avoid using "signature" when reported delta values

307: VPDB; report consistent precision (-5.28 and 0.40 or -5.3 and 0.4) – based on your methods these should be reported to one decimal.

309: Delta values, not isotopic signatures

313: VPDB

313, 314, 317: do not used "depleted"... the delta values are lower or higher

323-328: this method section seems a little incomplete compared to others...any citations on the method? What do you mean by "Mg-samples". How were samples dissolved and what kind of column chemistry was conducted to create solution that is loaded onto filaments?

335: "less radiogenic" is jargon....stick to the values and whether they are higher or lower than others...radiogenic can be used in a discussion when referring to a source rock or fluid, etc., but not when reporting the isotope ratios.

342: clumped isotope section lacks citations for the methods used

352, 353: mixing past and present tense

352: carbonate = calcite, dolomite, ?

375: replace "can be" with are or were depending on what tense you want to stick with

375: what are the errors on computed temperatures?

377: which carbonate is this fractionation factor for?

382: spell out 13

381: This results section is incomplete. The data is reported in table 3, but where are the actual clumped isotope data? I cannot find them in any of the materials, just the calculated temperatures. Please provide these. Also, the text only mentions the calculated ðİŽĚ18O value for the fluids, but do you want to also report in this text something about the oxygen vein isotope values? Perhaps this is covered prior to the methods for this section? This is an example of how the switching between methods and results is distracting to the reader.

388: include the "equation of Kim et al"

403: calcite should not be capitalized

405 – 409: The level of detail for this method of U-Pb dating is far briefer than your second method. Please provide equivalent levels of detail.

425: define "favorable U-Pb" levels

464/465: Are these the vein sets you said you were not going to discuss further on line 211?

470: same comment – is this repetitive from line 211 or different?

483: do not capitalize cardinal directions

484: what do you mean by abnormal burial

509/510: poorly worded and repetitive sentence

529: calcite? twins

533: histories

536: Is this sentence complete? It starts with a lower case letter

537: So, this is an example of where you need to provide a discussion of why a $\delta$18O value of 5 permil supports this interpretation. Just stating this as your interpretation is not good enough.

542: same comment

549: Sentence that starts with "That" does not make any sense.

556 – 570: This paragraph is really problematic from various standpoints. First, the writing is poor – pay attention to grammar and subject-verb agreement. The use of signature is overused. The use of depleted must be changed. Be consistent with VPDB and VSMOW.

564: "characterized by negative $\delta$18O" – so the negative sign is simply an artifact of the reference standard VSMoW. Why is a "negative" value significant here?

Interactive
comment

559 – 562: this seems contradictory as presented. . .How do "very high signatures of the fluids" correspond to "very depleted signatures" of the cements. Where is the calculation that shows this? I am looking at Figure 8 for some help here, and it seems you are overgeneralizing. From 110 – 140 C, your plot suggests both meteoric and "basinal contamination". . .cements from -17 to $\sim$-3, and fluids from $\sim$0 to 15 ‰

574 – 589: Again, pay attention to grammar. The prevalent usage of depleted, radiogenic, and signature need to be altered. Also, please provide discussion and citations for the claims that "very positive O isotope signatures" can be explained by your reasoning. This paragraph just needs reworking for clarification as it is really an integral part of your big interpretations.

590-594: This is a really long and confusing topic sentence.

594-595: consider using "difference in hydraulic head" rather than "water table height difference"

650: omit "unparalleled detail" – this is subjective to the reader

658: where are the "C isotope signatures" discussed? The $\delta$13C values are mentioned in the results and seemingly used in Figure 6, but what do they mean and how do they support the interpretations. Also where are the $\Delta$47 data?

666-667: this last sentence is true of course, but many studies have shown this. Consider a more impactful final sentence that highlight what your study has provided.

Figures

Fig 8: Justify the 0 ‰ division between meteoric and basinal in text and with citations. This is certainly not always the case. There are sedimentary basins with $\delta$18O values that are both greater and less than 0 ‰

Fig. 11. Please add numeric dates.

Table 2: Do not use signature in caption. These are delta values and sr isotope ratios.

Why the change in carbon isotope precision in the table?

Table 3. See prior comments on using "signature" and please provide the $\Delta 47$ data + errors.

[Figure]

---

## Referee Comment (RC2) · Anonymous Referee #2 · 9 Jun 2020

General Comments: This paper presents a multi-faceted approach to understanding the paleo-fluid history of the Tuscan Nappe region. A wide variety of data were used to fully evaluate the structural-fluid system, including: mineral vein petrography, oxygen and carbon stable isotopes, clumped isotopes, U-Pb dating of calcite veins, depth determination from stylolites, and Sr isotopes. The data appear to be good and the interpretations are generally supported by the data. All of the figures and tables are necessary. However, some of the data is not discussed in the text or presented in figures or tables.

[Figure]

Specific Comments: However, the paper was very difficult to read due to several is­sues: 1) The English grammar of the paper needs to be addressed, and be more consistent throughout. Some sections were well done, while others need work. In the detailed comments below, I address some of these, but a much more thorough review is needed. Also, maybe it was just my copy of the paper, but there were commonly no spaces after semi-colons in the reference lists.

2) The authors have explicitly not followed the standard paper format of methods, fol­lowed by results, followed by discussion. Instead, they have a method, followed by a brief results section, followed by another method, then another results section, and so on. This is very distracting, and the reader has to refocus on the results after each method. Similarly, the results are not fully discussed in their respective sections, and some results are in the methods sections, as well as in the discussion. I highly suggest that the authors redo in a more conventional format.

Detailed comments (by no means complete): Line 40: 'ubiquitous' seems like an odd word to use here. Maybe 'a variety'?

Line 47: instead of 'make', 'allow' or 'facilitate'

Line 52: instead of 'witness', maybe 'be influenced by'

Line 56: instead of 'for instance' use 'for example' as a lead off to the sentence.

Line 87: define 'LPS'

Line 91: 'is' not 'was' also elsewhere tense agreement.

Line 92: 'consider' not 'considered' (as above)

Line 100: be careful about using 'for the first time'. I suggest leaving it out.

Line 119: should be 'piggy-back'

Line 120 and throughout: please use dates in addition to stage/age names

[Figure]

Line 121: 'convexity' is awkward.

Line 128: what is meant by 'for long'? Line 134: maybe: 'Even if the interpretation of basement shortening s more accepted now'....

Line 138: 'In these last views' is awkward

Line 144: 'Nowadays' is awkward

Line 153-162: stage/age dates needed

Line 179-180: this is a result

Line 186: need parentheses around 'sigma 1'

Line 190: what is meant by 'regimes'?

Line 196: What are criteria for 'most representative fracture data'? and what about the other data?

Lines 210-212: the authors say that this set will not be discussed further, but it is covered later in lines 464-470.

Line 212: a better word than 'gathered' is needed

Line 255: give equation a number and take out of sentence.

Line 257: remove 'classic'

Line 284: instead of 'that filled up' use 'in'

Line 286: what are 'diagenetic states'?

Line 288: do not capitalize oxygen and carbon

Line 293: where were the host rock samples taken in proximity to the veins? Was there any vein-fluid host-rock interaction?

Line 298: how long were samples held at 90 degrees?

Line 307 and throughout: please be careful about using VPDB consistently.

Line 324: what are Mg-samples??

Line 330: instead of 'spread on' maybe use 'distributed over'

Lines 333 to 340: be careful of the use of radiogenic. Not appropriate here. Lines 352-367: need some references here on the methods used.

Lines 390-399: difficult to read, maybe present these data in table form.

Line 403: do not capitalize calcite

Lines: 405 to 409: need more details on the method here

Line 425: define 'favorable'

Line 445: instead of 'witnessing' maybe 'experiencing'

Line 478: instead of 'propose' use 'will'

Line 483: do not capitalize west

Line 484: what is 'abnormal burial'?

Lines 493-494: wording of this sentence is awkward

Lines 509-511: this sentence not clear

Line 529: specify 'calcite twins'

Line 533: instead of 'history' use 'histories'

Line 534: better explain 'clearly exhibit a singular history'

Line 536: is there something missing at the beginning of this paragraph?

Line 537: it is not clear how this value was obtained.

Line 542: same as above

Line 550: starting a sentence with 'That' is awkward

Line 554-570: this entire paragraph is very confusing and not clear. Please rewrite and also be consistent in VPDB.

Up to here, there has been no discussion of carbon values or the clumped isotopes.

Lines 574-577: careful of use of depleted, overprinted, 'very positive'

Lines 590-593: this is a very long sentence. Please break apart for clarity. Lines 594-595: this sentence is not clear. What is 'water table top difference in height'?

Line 650: remove 'unparalleled detail'

Comments of Figures and tables: Figure 3: should 'natural light' be 'plain polarized light'?

Figure 4 caption: what is meant by 'red color scale'?

Figure 6d: need 'VPDB'

Figure 7: placing formation seawater values on plot would be useful

Figure 8: contours are based on?

No figure or table with clumped isotope data.

VPDB is used throughout the paper, while V-PDB is used in the tables.

---

## Author Response (AR1)

**Dear Editor and Referees,**

We have now revised our manuscript. On behalf of all co-authors, I wish to thank the referees for their numerous comments and remarks, which are very similar to each other's. We did our best to comply with the remarks and comments of the referees, going beyond to make the manuscript better (changes tracked with blue highlights). On top of what will be reported in the specific answers, we also modified the geological map to reflect the regional distinction between the Tuscan units and the Umbria-Marche units. We also modified the presented burial model (changes highlighted with a red font), by taking into account the physical compaction due to burial (backstripping); and estimating the chemical compaction due to pressure solution from our dataset. While not modifying the results too much, it is an important step forward for this kind of approach and strengthen a lot the manuscript.

The following discussion reports the questions and remarks of the referees and our answers. In the line by line specific comments, we highlighted the question or comment in bold font, and answered it in regular font, referring to the line numbers of the new version of the manuscript. You will find the new highlighted version of the manuscript at the end of this document.

The manuscript is now 10163 word long from the title to the acknowledgements included, still comprises 11 figures, 3 tables, and a supplementary file.

**Anonymous Referee #1**

Received and published: 7 June 2020 The authors present an interpretation of the paleofluid flow history across the Tucscan Nappe, constrained by a multiproxy study. They integrate detailed structural analysis of stylolites fracture networks with isotopic (O and C stable isotope, Sr isotopes, clumped carbonate isotopes) and some limited U-Pb dating of calcite veins, cements, and fault coatings. Using the dates from this study in concert with other timing and burial constrains, the authors reconstruct a  $\sim$ 12 my history of fluid flow events that are linked to deformation. They identify evidence for both closed fluid systems and the open system flow of hydrothermal fluids.

This is an interesting and comprehensive evaluation of the role and evolution of fluids during regional tectonic compression. The interpretations appear mostly supported by the data, the data is of good quality, and the methods thorough in most cases. The data are high quality and this is certainly appropriate for publication in Solid Earth, and will be of interest to structural geologists, geochemists, and those working on problems in regional tectonics. One of the big contributions is the careful linkage between deformation and closed versus open-system fluid-rock interaction.

However, the reviewed version requires moderate revisions to improve the impact of the paper. I outline the major issues here, and then provide some line-by-line comments below.

 Writing: This paper needs a thorough technical edit paying particular attention to grammar, tense, subject-verb agreement, over-reliance on the passive voice, general clarity of sentences. Also, it still reads like it was prepared by multiple authors with different styles, and without careful review. In the line-by-line comments I point out many of the grammatical problems, but by no means are these comprehensive.

Author response: We agree the submitted version of the paper was prepared by several authors, and in spite of our efforts to make it seamless to read, results were below our expectation. The new version of the manuscript has been edited with much more care, following both's reviewers specific

comments and going further to ensure internal consistency on multiple levels, like style, results report, and organization.

2) Organization: The presentation style of method-results, method -results, etc. is quite distracting to this reviewer. The authors state in the paper that this is intentional with the goal of improving clarity. However, in order to follow all of the different and related data sets, it is very distracting to switch gears to reading another methods summary. True, I could skip over, but I would rather have the option of reading a single methods section, and a single results section. Also, this has resulted in some of the methods sections actually containing results, so they are merging together in some places. This is a relatively complicated data set to understand as a reader, so the organization needs to be strong.

Author response: The previous organization was the results of a bad choice of organization, the new version now separate clearly the methods from the results, and also the interpretation from the discussion. We believe that the classic organization methods-results-interpretation-discussion make the manuscript easier and clearer to read.

3) Over-reliance on "isotopic jargon": For clarity to non-experts and experts alike, it is important to use proper language when presenting and interpreting the isotopic results. For example, the stable isotope community prefers writing "δ18O values". Avoid use C2 of depleted, enriched, heavy, and light. . .be specific. If using enriched or depleted, please be specific as to what is enriched with respect to what. Similarly, for the Sr - isotopes. . ..avoid the "radiogenic" term unless being specific – for example, rather than writing that a sample is less radiogenic than another, just provide the values.. Also, the term "signature" is really overused in this paper. Be careful with its use. A stable isotope delta value is not a signature. . .avoid "δ18O signature of X ‰". It is OK to use phrases like "meteoric isotopic signature"

Author response: The form of the geochemical part of the paper has been edited to avoid using jargon, the term signature has been kept only at two places in the manuscript, where it make sense as it refers to the fluid reservoir and not to measurements. The other occurences were replaced by values. The term radiogenic has been removed as well, and values were provided and compared as suggested.

4) Adding numeric ages to Geologic Ages: For those of us who do not work in the Miocene, referring to the Late Aquitanian and Messinian time lacks clarity. Please include the numeric ages as well. One, this helps the reader piece together the geologic history of your study, and it teaches some of use more details of the time scale!

Author response: Thank you for this suggestion, we edited the text accordingly to add the absolute ages when we use a stage of the geological time scale.

5) Lack of adequate support/discussion for interpretations: The interpretation of the source fluids and degree of water-rock interaction are just stated. There is little discussion or justification and no examples from the literature. Please justify your interpretations. Also, the carbon isotope data is not discussed.

Author response: The fact we interpret the veins that show a d18O values higher than their host rock d18O values as witnessing an interaction between reservoir fluids and host rock is now extensively discussed in the manuscript. That interpretation also partly relies on the d13C values, that were not explicated in the previous version of the study, leading to confusion. Also, the wording "basinal

contamination" we used in that section was unfortunate, we meant that the precipitating fluids where formation fluids, evolving after various degrees of interaction with host rock, likely related to migrations through the basin. The complete text is located lines 558-589, but the specific case of interpreting our data as related to the degree of water rock interaction is a classic interpretation proposed since Clayton et al., 1966. It is now discussed line 581-589 as follow: "Indeed, if considering an environment with limited connection between reservoirs and no implication of external fluids, i.e. where fluids are sourced locally from the marine carbonates, the fact that in the western part of the UMAR, the  $\delta^{18}$ O values of the veins are significantly more positive than the  $\delta^{18}$ O values of their hostrocks implies that fluids that precipitated were enriched in 18O isotope relative to the original formational fluid. Such a fractionation is usually interpreted as the result of rock dissolution during fluid migration (Clayton et al., 1966; Hitchon and Friedman, 1969). This is further supported by the  $\delta^{18}O_{fluids}$  values derived from  $\Delta_{47}CO_2$  measurements, that are higher in the Monte Subasio (8 to 16‰ VSMOW, Fig. 8) than in the rest of the UMAR (from 0 to 8‰ VSMOW, Fig. 8), witnessing a higher degree of reservoir fluid-rock interaction in the western part of the UMAR."

6) U-Pb dating – this is outside of my expertise, but my read is that these veins only yielded limited information?

Author response: Indeed, as we stated in the manuscript – and emphasizes more in the new version (line 462-464)- veins did not hold enough U, or were too rich in Pb, for the ratio to be favourable to return an absolute age. Only the faults had a U/Pb ratio high enough to return an age. The fact that veins are less favourable for U/Pb dating in the UMAR is out of the scope of this study, but we refer to a recent review paper (Roberts et al., 2020) where such observation (not limited to UMAR) is tentatively discussed.

7) Clumped isotope data: The  $\Delta$ 47 results need to be added to Table 2 or 3, and presented in the results.

Author response: Thank you for pointing that out, the analytical results of D47CO2 are now reported with uncertainty in table 3 and are presented in the text from line 438 to 443. The corresponding d18O and d13C values were reported in Table 2 and plotted on Fig. 6. We want to highlight that during the review time, we added 3 more samples analysed with D47 CO2 in the Monte Catria (2) and in the Monte Conero (1), extending the across strike and along strike representativity of our dataset.

**line by line**

**40-48: Paragraph needs strengthening. . .currently a 2 sentence paragraph with a really long second sentence. Suggest breaking this down for more impact.**

We rephrased this paragraph which was too hard to read indeed, it is now line 61-78.

53: comma before and

Edited accordingly line 51

87: Define "LPS" here

Done line 85

90 – 91: mixing tense: are and was in same sentence

Edited line 87-89

100: This reviewer cautions the claim of "provides for the first time"...this is rarely true.

We removed this statement

**120-121: please provide numeric dates along with "Late Aquitanian"...etc. Also, should Late be capitalized?**

Edited accordingly, 'late' is not capitalized anymore and ages are provided all throughout the text (line 117)

**128: consider different and more active voice for "has been considered for long"**

We replaced by "UMAR was considered" (line 126)

**144: Replace "Nowadays" with Currently; "undergoes" with "is experiencing"**

Edited accordingly lines 141-142

**168 – 170: This reviewer finds this format difficult to follow.**

The manuscript has been reorganized.

**179-180: This is actually a result, not a method.**

Thank you for pointing that out, this sentence is now in the result section (line 360)

**186: need parentheses around sigma 1**

**Edited accordingly**

**197: How did you do the correction due to bedding dip – that is what program, or by hand.**

We edited the text to make this information available (line 189-194): "In order to capture the mesostructural and fluid flow evolution at the regional scale during layer-parallel shortening and folding, we gathered the statistically most representative fracture data by structure, regardless of the structural complexity in the individual folds, and corrected them from the local bedding dip using

an opensource stereodiagram rotation program (Grohmann and Campanha, 2010) to discriminate between early and syn-folding features."

**212: Can be gathered? consider different word choice**

**221: New paragraph at "Finally"**

For these two remarks, we reorganized this result section relative to the structural approach, in order to make it clearer.

**255: Perhaps a complicated equation like this should be a numbered one.**

OK, this is now equation (3)

**288: Oxygen and Carbon should not be capitalized**

We corrected accordingly

**293 – 303: need a method citation; specify dual inlet or continuous flow; probably not 105% or would crystallize. . .or did you measure density? How long were samples equilibrated at 90C?**

The protocol is a very standard protocol but I am afraid there is no dedicated publication for it with this machine at the SUERC. However we did edit the text to add the fact it was a dual inlet trap, and the duration of the sample equilibration at 90°C (30 min for calcite and 45 min for dolomite). The solution of H3PO4 is 103%, derived from density measurement indeed. (lines 257-268)

**306: avoid using "signature" when reported delta values**

Done here in everywhere in the text.

**307: VPDB; report consistent precision (-5.28 and 0.40 or -5.3 and 0.4) – based on your methods these should be reported to one decimal.**

Done here in everywhere in the text.

**309: Delta values, not isotopic signatures**

313: VPDB

Done here in everywhere in the text.

**313, 314, 317: do not used "depleted"... the delta values are lower or higher**

The new version does not use isotopic jargon such as depleted.

**323-328: this method section seems a little incomplete compared to others. . .any citations on the method? What do you mean by "Mg-samples". How were samples dissolved and what kind of column chemistry was conducted to create solution that is loaded onto filaments?**

This section has been edited completely and is much more complete now (lines 270-281). The Mg-samples was a mistake. The specific dissolution process is reported as follow (line 273 onwards) "The samples were then dissolved in ca. 1 ml of calibrated 2.5M HCl in preparation for column chemistry, and centrifuged. Samples were pipetted onto quartz-glass columns containing 4mls of AG50x8 cation exchange resin. Matrix elements were washed off the column using 48 ml of calibrated 2.5M HCl and discarded. Sr was collected in 12 ml of 2.5M HCl and evaporated to dryness."

335: "less radiogenic" is jargon. . ..stick to the values and whether they are higher or lower than others. . .radiogenic can be used in a discussion when referring to a source rock or fluid, etc., but not when reporting the isotope ratios.

Thank you, we do now stick to the values.

**342: clumped isotope section lacks citations for the methods used**

We now cite (line 290) as follow: "The clumped isotopes laboratory methods at Imperial College follow the protocol of Dale et al. (2014) as adapted for the automated clumped isotope measurement system IBEX (Imperial Batch EXtraction) system (Cruset et al., 2016)."

**352, 353: mixing past and present tense**

This part is consistent now, using past time (lines 292-294)

**352: carbonate = calcite, dolomite, ?**

Carbonate is calcite, no dolomite was used for clumped D47CO2 measurements (line 292).

**375: replace "can be" with are or were depending on what tense you want to stick with**

Done accordingly (line 313)

**375: what are the errors on computed temperatures?**

The 1 standard error reported in the table 3 is computed taking the following errors into consideration: the error in D47, the uncertainty in the Davies and John 2019 regression, and (for fluid) the error in d180 measured and the uncertainty in the calibration. In other words, the errors are properly propagated to your final temperature or d180fluid values and are reflected in the

uncertainties reported. I suggest you to refer to the Davies and John 2019 paper we cite for a more in depth discussion on the calibration itself.

**377: which carbonate is this fractionation factor for?**

It is for calcite (line 315)

**382: spell out 13**

Done (but for 16) line 436

381: This results section is incomplete. The data is reported in table 3, but where are the actual clumped isotope data? I cannot find them in any of the materials, just the calculated temperatures. Please provide these. Also, the text only mentions the calculated ð180 value for the fluids, but do you want to also report in this text something about the oxygen vein isotope values? Perhaps this is covered prior to the methods for this section? This is an example of how the switching between methods and results is distracting to the reader.

We did not report the D47CO2 results in the previous version, but they are now in the Table 3 and presented in the text line 438 onwards.

**388: include the "equation of Kim et al"**

We apologize for that was a mistake in citation, it is actually a fractionation equation from Kim and O'Neil, 1997, wich reads as follow  $1000 \ln(\alpha) = 18.030 \times \frac{10^3}{\tau} - 32.420$ . It is reported line 447.

**403: calcite should not be capitalized**

Edited accordingly

**405 – 409: The level of detail for this method of U-Pb dating is far briefer than your second method. Please provide equivalent levels of detail.**

Edited accordingly line 322-332

**425: define "favorable U-Pb" levels**

The term favourable is not strictly bound to a numerical value of the ratio. It is usually accepted the ratio should be 100 at minimum, but this depends heavily on the absolute content of U and Pb. We do not mean to provide a debatable and ad-hoc value in this paper. Please refer to Roberts and al., 2020, for an in-depth explanation.

**464/465: Are these the vein sets you said you were not going to discuss further on line 211?**

We apologize for the confusion, we edited the lines 365-366 (former line 211) and not refer to this vein sets anywhere further in the text.

**470: same comment – is this repetitive from line 211 or different?**

We remove this sentence.

**483: do not capitalize cardinal directions**

Edited accordingly

**484: what do you mean by abnormal burial**

We rephrased for the sake of clarity (line 521-522): "revealed that most of this unit locally underwent more burial because it was underthrusted below the Ligurian Nappe"

**509/510: poorly worded and repetitive sentence**

This part has been edited and reworded for the sake of clarity (line 547 onwards) : "We can therefore estimate an average duration of folding in the western-central part of the UMAR of ~3 Ma. Knowing the oldest record of post-orogenic extensional tectonics in the UMAR is mid-Pliocene (~3 Ma) (Barchi, 2010), we can also estimate the duration of the LSFT to ~2 Ma."

**529: calcite? twins**

Yes, it is now stated lines 566-569 : "Overall, most calcite grains from vein cements show thin twins (thickness < 5  $\mu$ m) and rectilinear, suggesting deformation at temperature below 170°C"

**533: histories**

This part was rewritten for the sake of clarity and to make it more convincing (line 570-607)

**536: Is this sentence complete? It starts with a lower case letter**

This part was rewritten for the sake of clarity and to make it more convincing (line 570-607)

**537: So, this is an example of where you need to provide a discussion of why a $\delta$ 180 value of 5 permil supports this interpretation. Just stating this as your interpretation is not good enough.**

This part was rewritten for the sake of clarity and to make it more convincing (line 570-607). We believe it is much clearer now that we don't rely our interpretation only on the value of d18Ofluids but on an array of data including Sr and 13C as well.

**542: same comment**

This part was rewritten for the sake of clarity and to make it more convincing (line 570-607). We believe it is much clearer now that we don't rely our interpretation only on the value of d180fluids but on an array of data including Sr and 13C as well.

**549: Sentence that starts with "That" does not make any sense.**

This part was rewritten for the sake of clarity

556 – 570: This paragraph is really problematic from various standpoints. First, the writing is poor – pay attention to grammar and subject-verb agreement. The use of signature is overused. The use of depleted must be changed. Be consistent with VPDB and VSMOW.

This part was rewritten for the sake of clarity and to make it more convincing (line 570-607

)

**564: "characterized by negative $\delta$ 180" – so the negative sign is simply an artifact of the reference standard VSMoW. Why is a "negative" value significant here?**

We agree "negative" is not the good wording, please refer to the new version of the paragraph (592-607).

**559 - 562: this seems contradictory as presented. . .How do "very high signatures of the fluids" correspond to "very depleted signatures" of the cements. Where is the calculation that shows this? I am looking at Figure 8 for some help here, and it seems you are overgeneralizing. From 110 - 140 C, your plot suggests both meteoric and "basinal contamination"...cements from - 17 to ~-3, and fluids from ~0 to 15 ‰**

Again, this paragraph suffered from wording issues, thanks to your comments, we managed to make it more clearer and now avoid the use of jargon. Also, "basinal contamination" was not the good wording for "degree of reservoir fluid-rock interaction". It was corrected here, in the whole text, and in the figures.

574 – 589: Again, pay attention to grammar. The prevalent usage of depleted, radiogenic, and signature need to be altered. Also, please provide discussion and citations for the claims that "very positive O isotope signatures" can be explained by your reasoning. This paragraph just needs reworking for clarification as it is really an integral part of your big interpretations.

This paragraph was edited

590-594: This is a really long and confusing topic sentence.

This paragraph was edited

**594-595: consider using "difference in hydraulic head" rather than "water table height difference"**

Done accordingly line 627

**650: omit "unparalleled detail" - this is subjective to the reader**

We removed that statement

658: where are the "C isotope signatures" discussed? The  $\delta$ 13C values are mentioned in the results and seemingly used in Figure 6, but what do they mean and how do they support the interpretations. Also where are the  $\Delta$ 47 data?

D13C is now discussed in the interpretation part relative to the fluid system, and a figure 6c has been added to present the d13C dataset, the D47 are now in the table 3 and described in the results section related to D47.

**666-667: this last sentence is true of course, but many studies have shown this. Consider a more impactful final sentence that highlight what your study has provided.**

We modified this last sentence to make it more specific to this study: "Beyond regional implications, the promising combination of stylolite roughness inversion and burial history reconstruction, linked to reliable past geothermal gradient appears as a powerful tool to unravel coupled structural and fluid flow evolution in fold-and-thrust belts."

**Figures**

Fig 8: Justify the 0 ‰ division between meteoric and basinal in text and with citations. This is certainly not always the case. There are sedimentary basins with  $\delta$ 180 values that are both greater and less than 0 ‰

We agree, the 'basinal' is not a good word for what we wanted to say, we replaced with "reservoir fluid-rock interaction", as explained in the text lines 586-588.

**Fig. 11. Please add numeric dates.**

Done

**Table 2: Do not use signature in caption. These are delta values and sr isotope ratios. Why the change in carbon isotope precision in the table?**

We amended the wording and modified the precision to reflect the uncertainty consistently in every table and in the text

Table 3. See prior comments on using "signature" and please provide the  $\Delta47$  data + errors.

Done.

**Anonymous Referee #2**

Received and published: 9 June 2020

General Comments: This paper presents a multi-faceted approach to understanding the paleofluid history of the Tuscan Nappe region. A wide variety of data were used to fully evaluate the structural-fluid system, including: mineral vein petrography, oxygen and carbon stable isotopes, clumped isotopes, U-Pb dating of calcite veins, depth determination from stylolites, and Sr isotopes. The data appear to be good and the interpretations are generally supported by the data. All of the figures and tables are necessary. However, some of the data is not discussed in the text or presented in figures or tables

Author response: We wish to thank the referee for this comment. In the new version all data are presented in the text, tables and figures.

Specific Comments: However, the paper was very difficult to read due to several issues:

 The English grammar of the paper needs to be addressed, and be more consistent throughout. Some sections were well done, while others need work. In the detailed comments below, I address some of these, but a much more thorough review is needed. Also, maybe it was just my copy of the paper, but there were commonly no spaces after semi-colons in the reference lists.

Author response: The text has been carefully edited to address grammar and style issues, following both reviewers comments and beyond. Spaces were added after semi-colons in the reference lists.

2) The authors have explicitly not followed the standard paper format of methods, followed by results, followed by discussion. Instead, they have a method, followed by a brief results section, followed by another method, then another results section, and so on. This is very distracting, and the reader has to refocus on the results after each method. Similarly, the results are not fully discussed in their respective sections, and some results are in the methods sections, as well as in the discussion. I highly suggest that the authors redo in a more conventional format.

Author response: We agree that the format we used was not adapted and led to confusion. In the new manuscript, we reorganized and separated the methods from the results, and the interpretation from the conclusion. We believe it results in a clearer text.

**Detailed comments (by no means complete):**

Line 40: 'ubiquitous' seems like an odd word to use here. Maybe 'a variety'?

We modified the sentence as follow (line 38): "The upper crust is the locus of omnipresent fluid migrations that occur at all scales"

**Line 47: instead of 'make', 'allow' or 'facilitate'**

We replaced by "facilitate" in line 45

**Line 52: instead of 'witness', maybe 'be influenced by'**

We replaced by "influence by", line 50

**Line 56: instead of 'for instance' use 'for example' as a lead off to the sentence.**

Edited accordingly, line 55

**Line 87: define 'LPS'**

Done, line 85

**Line 91: 'is' not 'was' also elsewhere tense agreement.**

Edited accordingly line 87-89

**Line 92: 'consider' not 'considered' (as above)**

Edited accordingly line 87-89

**Line 100: be careful about using 'for the first time'. I suggest leaving it out. We left that out.**

Line 119: should be 'piggy-back' Edited accordingly line 116

**Line 120 and throughout: please use dates in addition to stage/age names**

Edited accordingly, ages are provided all throughout the text (line 117)

Line 121: 'convexity' is awkward.

Edited and modified as "convex shape", line 119

**Line 128: what is meant by 'for long'?**

It was related to the long-standing debate over the subsurface structure of the ridge, we left that wording out to avoid confusion.

**Line 134: maybe: 'Even if the interpretation of basement shortening s more accepted now'....**

Done line 131

**Line 138: 'In these last views' is awkward**

Edited as "For the latter interpretations" (line 136)

**Line 144: 'Nowadays' is awkward**

This has been replaced by "currently" (line 141)

**Line 153-162: stage/age dates needed**

Done from line 150 onwards

**Line 179-180: this is a result**

Thank you for pointing that out, this sentence is now in the result section (line 360)

**Line 186: need parentheses around 'sigma 1'**

Edited accordingly

**Line 190: what is meant by 'regimes'?**

We know clearly say what the stress regime can be (extensional, reverse, strike slip) line 185.

**Line 196: What are criteria for 'most representative fracture data'? and what about the other data?**

The representativity of the veins is a statistical criterion given by Fisher statistical test. We edited the text as "we gathered the statistically most representative fracture data" (line 191). The other data are simply not first order and can also relate to local structural complexity. That is what we say line. 192.

**Lines 210-212: the authors say that this set will not be discussed further, but it is covered later in lines 464-470.**

We removed the part line 464-470 and do not discuss this set later on.

**Line 212: a better word than 'gathered' is needed**

This paragraph was edited completely. We believe it to be clearer now.

**Line 255: give equation a number and take out of sentence.**

OK, this is now equation (3)

**Line 257: remove 'classic'**

Done

**Line 284: instead of 'that filled up' use 'in'**

Done line 248

**Line 286: what are 'diagenetic states'?**

We rephrased as follow (line 250-251): "possible post-cementation diagenesis such as dissolution or replacement were checked under cathodoluminescence"

**Line 288: do not capitalize oxygen and carbon**

**Edited accordingly**

**Line 293: where were the host rock samples taken in proximity to the veins? Was there any vein-fluid host-rock interaction?**

Host rocks where sampled on the same hand-cut, about 2 cm from veins. It is now explicated in the text line 258.

**Line 298: how long were samples held at 90 degrees?**

30 minutes for calcite and 45 minutes for dolomite, as now stated line 263.

**Line 307 and throughout: please be careful about using VPDB consistently.**

This is now VPDB everywhere in text, table and figures.

**Line 324: what are Mg-samples??**

The paragraph about the methodology of Sr was completely edited. Mg-samples was a mistake.

**Line 330: instead of 'spread on' maybe use 'distributed over'**

Edited accordingly line 423.

**Lines 333 to 340: be careful of the use of radiogenic. Not appropriate here.**

We removed the term radiogenic from the paper as it was not adapted.

**Lines 352-367: need some references here on the methods used.**

We now cite (line 290) as follow: "The clumped isotopes laboratory methods at Imperial College follow the protocol of Dale et al. (2014) as adapted for the automated clumped isotope measurement system IBEX (Imperial Batch EXtraction) system (Cruset et al., 2016)."

**Lines 390-399: difficult to read, maybe present these data in table form.**

We understand that it is not easy to read, but as we need to present the results, we don't have a lot of choice. They are already reported in table 3.

**Line 403: do not capitalize calcite**

Edited accordingly.

**Lines: 405 to 409: need more details on the method here**

We developed this method part line 322-332.

**Line 425: define 'favorable'**

The term favourable is not strictly bound to a numerical value of the ratio. It is usually accepted the ratio should be 100 at minimum, but this depends heavily on the absolute content of U and Pb. We do not mean to provide a debatable and ad-hoc value in this paper. Please refer to Roberts and al., 2020, for an in-depth explanation.

**Line 445: instead of 'witnessing' maybe 'experiencing'**

The form of this part was edited

**Line 478: instead of 'propose' use 'will'**

Edited accordingly line 510

**Line 483: do not capitalize west**

Edited accordingly

**Line 484: what is 'abnormal burial'?**

We rephrased for the sake of clarity (line 521-522): "revealed that most of this unit locally underwent more burial because it was underthrusted below the Ligurian Nappe"

**Lines 493-494: wording of this sentence is awkward**

We rephrased as (line 530-534): "In the case of the UMAR, 800m is the minimum depth at which dissolution stopped along BPS planes, regardless of studied formations. That confirms that burial-related pressure solution (i.e., chemical vertical compaction) likely initiated at depth shallower than 800 m (Ebner et al., 2009b; Rolland et al., 2014; Beaudoin et al., 2019; Beaudoin et al., 2020). "

**Lines 509-511: this sentence not clear**

This part has been edited and reworded for the sake of clarity (line 547 onwards) : "We can therefore estimate an average duration of folding in the western-central part of the UMAR of ~3 Ma. Knowing the oldest record of post-orogenic extensional tectonics in the UMAR is mid-Pliocene (~3 Ma) (Barchi, 2010), we can also estimate the duration of the LSFT to ~2 Ma."

**Line 529: specify 'calcite twins'**

Edited accordingly line 566

**Line 533: instead of 'history' use 'histories'**

This part was rewritten for the sake of clarity and to make it more convincing (line 570-607)

**Line 534: better explain 'clearly exhibit a singular history'**

This part was rewritten for the sake of clarity and to make it more convincing (line 570-607)

**Line 536: is there something missing at the beginning of this paragraph?**

This part was rewritten for the sake of clarity and to make it more convincing (line 570-607)

**Line 537: it is not clear how this value was obtained.**

This part was rewritten for the sake of clarity and to make it more convincing (line 570-607). We believe it is much clearer now that we don't rely our interpretation only on the value of d180fluids but on an array of data including Sr and 13C as well.

**Line 542: same as above**

This part was rewritten for the sake of clarity and to make it more convincing (line 570-607). We believe it is much clearer now that we don't rely our interpretation only on the value of d180fluids but on an array of data including Sr and 13C as well.

**Line 550: starting a sentence with 'That' is awkward**

This part was rewritten for the sake of clarity

**Line 554-570: this entire paragraph is very confusing and not clear. Please rewrite and also be consistent in VPDB. Up to here, there has been no discussion of carbon values or the clumped isotopes.**

This part was rewritten for the sake of clarity and to make it more convincing (line 570-607). Carbon isotopes are now represented on Fig. 6c, and discussed in this paragraph as an important part that our interpretation is built on. (lines 570-580)

**Lines 574-577: careful of use of depleted, overprinted, 'very positive'**

This part was rewritten with no jargon.

**Lines 590-593: this is a very long sentence. Please break apart for clarity.**

Done line 622-625.

**Lines 594-595: this sentence is not clear. What is 'water table top difference in height'?**

It is the difference in the water head, we edited to make it clearer line 627.

**Line 650: remove 'unparalleled detail'**

OK.

**Comments of Figures and tables:**

**Figure 3: should 'natural light' be 'plain polarized light'?**

Indeed, thank you for pointing that out, it stands corrected.

**Figure 4 caption: what is meant by 'red color scale'?**

We meant that the red areas on the stereograms indicates the highest fracture pole densities. We reworded to make it clear.

**Figure 6d: need 'VPDB'**

Done accordingly

**Figure 7: placing formation seawater values on plot would be useful**

The formation seawater values are given by the host rock values, it is indicated now in the caption.

**Figure 8: contours are based on?**

Contours are based on uncertainties, as stated now in the caption.

**No figure or table with clumped isotope data**

It is now in reported in table 3

**VPDB is used throughout the paper, while V-PDB is used in the tables**

Table have been amended.

Regional-scale paleofluid system across the Tuscan Nappe - Umbria Marche Apennine Ridge (northern Apennines) as revealed by mesostructural and isotopic analyses of stylolite-vein networks

Nicolas Beaudoin1,\*, Aurélie Labeur1,2, Olivier Lacombe2, Daniel Koehn3, Andrea Billi4, Guilhem Hoareau1, Adrian Boyce5, Cédric M. John7, Marta Marchegiano7, Nick M. Roberts6, Ian L. Millar6, Fanny Claverie8, Christophe Pecheyran8, and Jean-Paul Callot1.

- 1. Universite de Pau et des Pays de l'Adour, E2S UPPA, LFCR, Pau, France (nicolas.beaudoin@univ-pau.fr)
- 2. Sorbonne Université, CNRS-INSU, Institut des Sciences de la Terre de Paris ISTeP, Paris, France
- 3. GeoZentrum Nordbayern, University Erlangen-Nuremberg, Erlangen, Germany
- 4. Consiglio Nationale delle Ricerche, Roma, Italy
- 5. Scottish Universities Environmental Research Centre (SUERC), East Killbride, UK
- 6. Geochronology and Tracers Facility, British Geological Survey, Environmental Science Centre, Nottingham, NG12 5GG, UK
- 7. Department of Earth Sciences & Engineering, Imperial College London, London, UK
- 8. Universite de Pau et des Pays de l'Adour, E2S UPPA, IPREM, Pau, France

**Abstract**

We report the results of a multi-proxy study that combines structural analysis of a fracture-stylolite network and isotopic characterization of calcite vein cements/fault coating. Together with new paleopiezometric and radiometric constraints on burial evolution and deformation timing, these results provide a first-order picture of the regional fluid systems and pathways that were present during the main stages of contraction in the Tuscan Nappe and Umbria Marche apennine ridge (Northern Apennines). We reconstruct four steps of deformation at the scale of the belt: burial-related stylolitization, Apenninic-related layer-parallel shortening with a contraction trending NE-SW, local extension related to folding and late stage fold tightening under a contraction still striking NE-SW. We combine the paleopiezometric inversion of the roughness of sedimentary stylolites - that constrains the range of burial depth of strata prior to layer-parallel shortening - with burial models and U-Pb absolute dating of fault coatings in order to determine the timing of development of mesostructures. In the western part of the ridge, layer-parallel shortening started in Langhian time (~15 Ma), then folding started at Tortonian time (~8 Ma), late stage fold tightening had started by the early Pliocene (~5 Ma) and likely lasted until recent/modern extension occurred (~3 Ma onward). The textural and geochemical ( $\delta^{18}$ O,  $\delta^{13}$ C,  $\Delta_{47}$ CO2 and  ${}^{87}$ Sr/ ${}^{86}$ Sr) study of calcite vein cements and fault coatings reveals that most of the fluids involved in the belt during deformation are either local or flowed laterally from the same reservoir. However, the western edge of the ridge recorded pulses of eastward squeegeetype migration of hydrothermal fluids (>140°C), that can be related to the difference in structural style of the subsurface between the eastern Tuscan Nappe and the Umbria Marche Ridge.

**Introduction**

The upper crust is the locus of omnipresent fluid migrations that occur at all scales, leading to strain localization, earthquake triggering and georesource generation, distribution and storage (e.g., Cartwright, 2007; Andresen, 2012; Bjørlykke, 1994, 1993; Lacombe and Rolland, 2016; Lacombe et al., 2014; Roure et al., 2005; Agosta et al., 2016). Carbonate rocks host an important part of the world's exploited hydrocarbons, strategic ores and water resources (Agosta et al., 2010). It is thus a

fundamental topic to depict the history of fluid migration in deformed carbonates. Such knowledge impacts both the prediction and monitoring of energy prospect and potential storage area, but also may help refine our understanding of what mechanisms **facilitate** fluid migrations during diagenesis of the sedimentary rocks, along with the time and spatial scales of fluid flow.

Fluid migration events and related accumulations are usually linked to past tectonic events, especially to the large-scale faults and fracture networks created during these tectonic events. Indeed, structural studies established that fracture networks in folded reservoirs are not exclusively related to the local folding history (Stearns and Friedman, 1972), and can also be influenced by burial history (Becker et al., 2010; Laubach et al., 2010; Laubach et al., 2019), and long-term and large-scale regional deformation (Lacombe et al., 2011; Quintà and Tavani, 2012; Tavani and Cifelli, 2010; Tavani et al., 2015; Bellahsen et al., 2006; Bergbauer and Pollard, 2004; Ahmadhadi et al., 2008; Sassi et al., 2012; Beaudoin et al., 2012; Amrouch et al., 2010). In fold-and-thrust belts and orogenic forelands, it is for example possible to subdivide the mesoscale deformation (faults, veins, stylolites) history into specific stages (Tavani et al., 2015): extension related to foreland flexure and bulging; pre-folding layer-parallel shortening (kinematically unrelated with folding); early folding layer-parallel shortening; syn-folding, strata curvature-related, local extension; late stage fold tightening, the last three stages being kinematically related with folding; and post-folding contraction or extension (kinematically unrelated with folding).

In the past decades, a significant volume of work has been conducted in order to reconstruct past fluid migrations through either localized fault systems or distributed sub-seismic fracture networks, in relation with past tectonic events, from the scale of a single fold to that of the basin itself (Engelder, 1984; Reynolds and Lister, 1987; McCaig, 1988; Evans et al., 2010; Evans and Hobbs, 2003; Evans and Fischer, 2012; Forster and Evans, 1991; Cruset et al., 2018; Lacroix et al., 2011; Travé et al., 2000; Travé et al., 2007; Bjørlykke, 2010; Callot et al., 2017a; Callot et al., 2017b; Roure et al., 2010; Roure et al., 2005; Van Geet et al., 2002; Vandeginste et al., 2012; Vilasi et al., 2009; Barbier et al., 2012; Beaudoin et al., 2011; Beaudoin et al., 2014; Beaudoin et al., 2013; Beaudoin et al., 2015; Fischer et al., 2009; Lefticariu et al., 2005; Di Naccio et al., 2005). A variety of fluid system evolution arises from published studies, and it is established that fracture networks and related mineralization can witness long term fluid migration in folded reservoirs. In some cases, fluid migration is stratigraphically compartmentalized and directed by compressive tectonic stress. In other cases, mineralization record an infill of meteoric fluids flowing downward, or of hydrothermal fluids (*i.e.* hotter than the host-rock they precipitated in) flowing upward either from the basin or the basement rocks through large scale faults or décollement levels (
[revised manuscript text omitted]

Sedimentary stylolites also yield quantitative information on the volume of dissolved rocks during burial (Toussaint et al., 2018), the minimum of which can be approached in 1D by measuring the amplitude of the highest peak along the stylolite track (Table 1), and multiplying this height value (in m) by the average density of stylolites (#/m) derived from field spacing measurement (Fig. 2 c-d, in m).

**c. O, C stable isotopes**

Calcite cements in tectonic veins related either to layer-parallel shortening or to strata curvature at fold hinges were studied petrographically (Fig. 3). The vein textures were characterized in thin sections under an optical microscope, and possible post-cementation diagenesis such as dissolution or replacement were checked under cathodoluminescence, using a cathodoluminescence CITL CCL 8200 Mk4 operating under constant gun condition of 15kV and 300µA. To perform oxygen and carbon stable isotope analysis on the cements that were the most likely to witness the conditions of fluid precipitation at the time the veins opened, we selected those veins that (1) show no obvious

evidence of shear; (2) the texture of which was elongated blocky or fibrous (Fig. 3); and (3) show homogeneous cement under cathodoluminescence (Fig. 3), precluding any later diagenetic alteration.

40 µg of calcite powder was hand sampled for each of 58 veins and 54 corresponding hostrocks (sampled ~2 cm away from veins) in various structures and formation along the transect, in both TN and UMAR. Carbon and oxygen stable isotopes were analyzed at the Scottish Universities Environmental Research Centre (SUERC, East Kilbride, UK) on an Analytical Precision AP2003 dual inlet mass spectrometer equipped with a separate acid injector system. As samples were either pure calcite or pure dolomite, we placed samples in glass vials to conduct a reaction with 103% H3PO4 under a helium atmosphere at 90°C, for 30 and 45 min on calcite and dolomite, respectively. Results are reported in table 2, in permil relative to Vienna PeeDee Belemnite (% VPDB). Mean analytical reproducibility based on replicates of the SUERC laboratory standard MAB-2 (Carrara Marble) was around  $\pm$  0.2‰ for both carbon and oxygen. MAB-2 is an internal standard extracted from the same Carrara Marble quarry, as is the IAEA-CO208 1 international standard. It is calibrated against IAEA-CO-1 and NBS-19.

**d. 87Sr /86Sr measurements**

Sr-isotope analysis was performed at the Geochronology and Tracers Facility, British Geological Survey. 2-10 mg of sample was weighed into 15ml Savillex teflon beakers and dissolved in 1-2mls of 10% Romil uPA acetic acid. After evaporating to dryness, the samples were converted to chloride form using 2 ml of Teflon-distilled HCI. The samples were then dissolved in ca. 1 ml of calibrated 2.5M HCl in preparation for column chemistry, and centrifuged. Samples were pipetted onto quartz-glass columns containing 4mls of AG50x8 cation exchange resin. Matrix elements were washed off the column using 48 ml of calibrated 2.5M HCl and discarded. Sr was collected in 12 ml of 2.5M HCl and evaporated to dryness.

Sr fractions were loaded onto outgassed single Re filaments using a TaO activator solution and analyzed in a Thermo-Electron Triton mass spectrometer in multi-dynamic mode. Data are normalized to 86Sr/88Sr = 0.1194. Three analyses of the NBS987 standard run with the samples gave a value of 0.710250 ± 0.000001 (1-sigma).

**e. Carbonate clumped isotope paleothermometry ( $\Delta_{47}$ CO2)**

Clumped isotopes analyses were carried out in the Qatar Stable Isotope Laboratory at Imperial College London. The technique relies on the tendency for heavy isotopes (13C, 18O) to 'clump' together in the same carbonate molecule, that varies only by temperature. Since the clumping of heavy isotopes within a molecule is a purely stochastic process at high temperature but is systematically over-represented (relative to randomly distributing isotopes among molecules) at low temperature, the 'absolute' temperature of carbonate precipitation can be constrained using clumped isotope abundances.

The clumped isotopes laboratory methods at Imperial College follow the protocol of Dale et al. (2014) as adapted for the automated clumped isotope measurement system IBEX (Imperial Batch EXtraction) system (Cruset et al., 2016). Typical sample size was 3.5 mg of calcite powder per replicate. Measurement of 13C-18O ordering in sample calcite was achieved by measurement of the relative abundance of the 13C18O16O isotopologues (mass 47) in acid evolved CO2 and is referred in this paper as  $\Delta_{47}$  CO2. A single run on the IBEX comprises 40 analyses, 30% of which are standards. Each analysis takes about 2 hours. The process starts with 10 minutes of reaction of the carbonate powder in a common acid bath containing 105% orthophosphoric acid at 90°C to liberate CO2. The CO2 gas is then captured in a water/CO2 trap maintained at liquid nitrogen temperature, and then moved through a hydrocarbon trap filled with poropak and a second water trap using helium as carrier gas. At the end of the cleaning process, the gas is transferred into a cold finger attached to the mass spectrometer, and into the bellows of the mass spectrometer. Following transfer, analyte CO2 was measured on a dual inlet Thermo MAT 253 mass spectrometers (MS "Pinta"). The reference gas used is a high purity CO2, with the following reference values: -37.07‰  $\delta^{13}C_{VPDB}$ , 8.9‰  $\delta^{18}O_{VSMOW}$ . Measurements comprise 8 acquisitions each with 7 cycles with 26s integration time. A typical acquisition time is 20 minutes, corresponding to a total analysis time of 2 hours.

Data processing was carried out in the freely available stable isotope management software, *Easotope* (www.easotope.org, John and Bowen, 2016). The raw  $\Delta_{47}$  CO2 is corrected in three steps. First, mass spectrometer non-linearity was corrected by applying a "pressure baseline correction" (Bernasconi et al., 2013). Next, the  $\Delta_{47}$  results were projected in the absolute reference frame or Carbon Dioxide Equilibrated Scale (CDES, Dennis et al., 2011) based on routinely measured ETH1, ETH2, ETH3, ETH4 and Carrara Marble (ICM) carbonate standards (Meckler et al., 2014; Muller et al., 2017). The last correction to the raw  $\Delta_{47}$  was to add an acid correction factor of 0.082‰ to obtain a final  $\Delta_{47}$ CO2 value (Defliese et al., 2015). Temperatures of precipitation were then estimated using the equation of Davies and John (2019). The bulk isotopic value of  $\delta^{18}$ O was corrected for acid digestion at 90°C by multiplying the value by 1.0081 using the published fractionation factor valid for calcite (Kim et al., 2007). Contamination was monitored by observing the values on mass 48 and 49 from each measurement, using a  $\Delta_{48}$  offset value > 0.5‰ and/or a 49 parameter values > 0.3 as a threshold to exclude individual replicates from the analysis (Davies and John, 2019).

f. U-Pb absolute dating of veins and faults

The calcite U-Pb geochronology was conducted in two different ways, of which specific methodology is reported as supplementary material:

- LA-ICPMS trace elements and U-Pb isotope mapping were performed at the Geochronology and Tracers Facility, British Geological Survey, UK, on 6 vein samples. Data were generated using a Nu Instruments Attom single collector inductively coupled plasma mass spectrometer coupled to a NWR193UC laser ablation system fitted with a TV2 cell, following protocol reported previously (Roberts et al., 2017; Roberts and Walker, 2016). Laser parameters were 110 µm spots, ablated at 10 Hz for 30 seconds with a fluence of 7 J/cm2. WC1 (Roberts et al., 2017) was used as a primary reference material for Pb/U ratios, and NIST614 for Pb/Pb ratios; no secondary reference materials were run during the session. Additional constraints on U-Pb composition were calculated from the Pb and U masses measured during the trace element mapping. Baselines were subtracted in Iolite, and Pb/Pb and Pb/U ratios were calculated offline in excel. No normalisation was conducted, as the raw ratios are suitable accurate to assess

- LA-ICPMS U-Pb isotope mapping approach was undertaken at the Institut des Sciences Analytiques et de Physico-Chimie pour l'Environnement et les Matériaux (IPREM) Laboratory (Pau, France). All the 29 samples were analysed with a 257 nm femtosecond laser ablation system (Lambda3, Nexeya, Bordeaux, France) coupled to an HR-ICPMS Element XR (ThermoFisher Scientific, Bremen, Germany) fitted with the Jet Interface (Donard et al., 2015). The method is based on the construction of isotopic maps of the elements of interest for dating (U,Pb,Th) from ablation along lines, with ages calculated from the pixel values (Hoareau et al., 2020). The ablation was made in a helium atmosphere (600 mL

min-1), and 10 mL min-1 of nitrogen was added to the helium flow before mixing with argon in the ICPMS. Measured wash out time of the ablation cell was  $\sim$ 500 ms for helium gas. The fs-LA-ICP-MS coupling was tuned daily, and the additional Ar carrier gas flow rate, torch position and power were adjusted so that the U/Th ratio was close to 1 +/- 0.05 when ablating the glass SRM NIST612. Detector cross-calibration and mass bias calibration were checked daily. The laser and HR-ICPMS parameters used for U-Pb dating are detailed in the supplementary material.

**3. Results**

a. Mesostructural analysis of joints, veins and striated fault planes

Based on the average orientation and the angle to the local fold axis, we group veins/joints in 2 main sets labelled J (Fig. 4):

- set J1 comprises joints/veins at high angle to bedding, that strike E-W to NE-SW but perpendicular to the local strike of fold axis. The trend of this set J1 evolves eastward as follows: E-W in the westernmost part (Cetona, Subasio), E-W to NE-SW in the central part (Catria, Nero), NE-SW in the eastern part of the chain (San Vicino, Cingoli), and ENE-WSW in the far foreland (Conero).

- set J2 comprises joints/veins at high angle to bedding that strike parallel to the local trend of the fold hinge, *i.e.* NW-SE in the ridge to N-S in the outermost part of the belt, where the arcuate shape is more marked.

Note that as set J1 strikes perpendicular to the local strata direction, it is impossible to infer a pre-, syn- or post-tilting (post-tilting then called J3 hereinafter) origin for its development. In most case though, abutment relationships establish a relative chronology with set J1 predating set J2 (Fig. 3). The veins of sets J1 and J2 show twinned calcite grains (Fig 3) with mostly thin and rectilinear twins (thickness < 5  $\mu$ m, Fig. 3). Another set comprising joints striking N-S and oblique to the direction of the fold axis is documented in the Monte Catria. It is also encountered at other locations but can be regarded as a second order set at regional scale on a statistical basis. This set could be tentatively related to lithospheric flexure (Mazzoli et al., 2002; Tavani et al., 2012), but as it is the least represented in our data, it will not be considered hereinafter.

Most tectonic stylolites have peaks striking NE-SW (Fig. 4), but they can be split in two sets labelled S based on the orientation of their planes with respect to the local bedding:

- Set S1 comprises bed-perpendicular, vertical stylolite planes containing horizontal peaks in the unfolded attitude of strata;
- Set S2 comprises ~vertical stylolite planes containing ~horizontal peaks in the current attitude of strata (set S2).

S1 and S2 are not always easily distinguished when both occurred at the fold scale because (1) stylolite data were often collected in shallow dipping strata, (2) peaks are not always perpendicular to the stylolite planes, and (3) the orientation data are scattered with intermediate plunges of the peaks. Another set showing stylolite planes with N-S peaks parallel to bedding is documented at Monte Subasio only, thus will not be considered in the regional sequence of deformation.

Finally, some mesoscale reverse and strike-slip conjugate fault systems have been measured (sets labelled F), of which fault-slip data inversion under specific assumptions (e.g., Lacombe, 2012) yields (1) a NE-SW contraction in the unfolded attitude of the strata (early folding set F1, flexural-slip related, bedding-parallel reverse faults) and (2) a NE-SW contraction in the current attitude of the strata (late folding set F2, strike-slip conjugate faults and reverse faults).

**b. Inversion of sedimentary stylolites**

The paleopiezometric study of 30 bedding-parallel stylolites returned a range of burial depths, across the UMAR, from W to E, reported in table 1. Most data come from the western part of the UMAR: in the Subasio Anticline (n=7), the depth returned by the Scaglia Bianca and the lower part of the Scaglia Rossa Fms. ranges from ca.  $800 \pm 100$  m to ca.  $1450 \pm 150$  m. In Fiastra area (n=6), the depth returned for the Maiolica Fm. ranges from  $800 \pm 100$  m to  $1200 \pm 150$  m. In the Gubbio fault area (n=4), the depth returned for the Jurassic Corniola Fm. ranging from  $600 \pm 70$  m to  $1450 \pm 150$  m. In the Gubbio fault area (n=4), the depth returned for the Jurassic Corniola Fm. ranging from  $600 \pm 70$  m to  $1450 \pm 150$  m. In the Monte Nero (n=11), the depth data published by Beaudoin et al., (2016), and updated here range from 750 ± 100 m to 1350 ± 150 m in the Maiolica. Fewer data comes from the western part of the UMAR: in the Monte San Vicino (n=2), the depth returned for the Iower part of the Scaglia Rossa is  $650 \pm 70$  m in the foreland at Conero Anticline (n=1). The maximum height of peaks along the studied stylolite tracks ranges from 0.6 to 8.5 mm (n=30) with a mean value of 2.6 mm. Spacing values for these stylolites were measured on outcrops (Fig. 2) and range from 1 to 2 cm, averaging the number of stylolite per meter to 70. Considering that dissolution is isotropic along the stylolite plane, the volume of rock loss in relation to the chemical compaction is 18%.

**c. O, C stable isotopes**

At the scale of the study area, most formations cropping out were sampled (Table 2), and oxygen isotopic values of the vein cements and striated fault coatings range from -16.8‰ to 3.7‰ VPDB while in the host rocks values range from -5.3 ‰ to 0.4‰ VPDB. Carbon isotopic values range from -9.7‰ to 2.7‰ VPDB, and from 0.0‰ to 3.5‰ VPDB in the veins and in the host rock, respectively (Fig. 6a-b). Isotopic values are represented either according to the structure where they have been sampled in, irrespective of the structural position in the structure (*i.e.* limbs or hinge), or according to the set they belong to, differentiating the sets J1, J2 and F1 (Fig. 6a). At the scale of the belt, isotopic values of host rocks are very similar, the only noteworthy point being that the Triassic carbonates have lower  $\delta^{18}$ O than the rest of the column ( $\delta^{18}$ O of -5.5% to -3.5% versus -3.2% to -1.0% VPDB, Fig. 6b). Considering the vein cements, the  $\delta^{13}$ C values are rather similar in all structures and in all sets, a vast majority of veins showing cements with values of 1.5 ±1.5‰ VPDB. In most structures, the  $\delta^{13}$ C values of the veins are similar to the  $\delta^{13}$ C values of the host rock, with the notable exception of the veins hosted in Triassic carbonates of the Cetona anticline, where the shift between  $\delta^{13}$ C values of veins and  $\delta^{13}$ C values of host rocks ranges from +4.0‰ VPDB to +7.5‰ VPDB (Fig. 6c). The  $\delta^{18}$ O values are ranging from -6.0% VPDB to +3.7% VPDB in most of the structures and formation, irrespectively of vein set. However, veins sampled in Monte Subasio and Monte Corona return very negative  $\delta^{18}$ O values < -15.0‰ VPDB (Fig. 6a). The shift between the  $\delta^{18}$ O value of the vein and the  $\delta^{18}$ O value of the surrounding host rock (Fig. 6d) increases from the western part of the belt (down to -15.0‰ VPDB in Monte Corona and Monte Subasio) to the central and eastern part of the belt (up to +5.0‰ VPDB in Monte Nero, Monte San Vicino, Monte Conero).

**d. 87Sr/86Sr measurements**

Analyses were carried out on 7 veins and 6 corresponding host rocks, distributed over three structures of the UMAR (Monte Subasio, Monte Nero and Monte San Vicino, from the hinterland to the foreland) and three formations (the Calcare Massiccio, the Maiolica, and the Scaglia Fms., Fig. 7, Table 2). Vein sets sampled are the J1, J2 and J3 sets described in the whole area. 87Sr/86Sr values of host rocks differ according to the formations, being the highest in the Scaglia Rossa Fm. (87Sr/86Sr  $\approx$  0.70780), intermediate in the Calcare Massicio Fm. (87Sr/86Sr  $\approx$ 0.70760), and the lowest in both the Scaglia Bianca and the Maiolica Fm. (87Sr/86Sr  $\approx$  0.70730). 87Sr/86Sr values of host rocks are in line with expected values for seawater at the time of their respective deposition (McArthur et al., 2001). 87Sr/86Sr values of veins scatter from 0.70740 to 0.70770, with lower values in the Monte Nero and in the Monte Subasio (0.707644 to 0.707690, Set J2). One vein cement of J3 in the Monte Subasio returned a lower 87Sr/86Sr value of 0.707437.

**e. Carbonate clumped isotope paleothermometry ( $\Delta_{47}$ CO2)**

Sixteen samples were analyzed, including cements of NE-SW (J1) and NW-SE (J2) pre-folding vein sets, along with coatings of early folding reverse (F1) and late folding strike-slip conjugate mesoscale faults (F2). Regardless of the structural position in the individual folds,  $\Delta_{47}CO_2$  values for veins (Table 3) range from 0.511 ±0.004 to 0.608 ±0.000‰CDES in the Monte Corona (n=3), from 0.468 ±0.032 to 0.574 ±0.000‰CDES in the Monte Subasio (n=5), from 0.662 ±0.004 to 0.685 ±0.052 ‰CDES in the Monte Nero (n=2), from 0.568 ±0.035 to 0.658 ±0.018 ‰CDES in the Monte San Vicino and the syncline to its west (n=4), from 0.601 ±0.032 to 0.637 ±0.013 ‰CDES in the Monte Catria (n=2) and is of 0.643 ±0.044 ‰CDES in the Monte Conero. Corresponding  $\delta^{18}$ O and  $\delta^{13}$ C are reported in Table 2. Analysis of  $\Delta_{47}CO_2$  returns the precipitation temperature (T) and the oxygen isotopic values of the mineralizing fluid can be calculated using the  $\delta^{18}$ O of the mineral, the clumped isotope temperature and the fractionation equation providing the fractionation coefficient lpha as a function of the temperature T:  $1000 \ln(\alpha) = 18.030 \times \frac{10^3}{T} - 32.420$  (Kim and O'Neil, 1997) (Fig. 8). Veins and faults belong to the Calcare Massiccio Fm., the Maiolica Fm., the Scaglia Fm., and the marls of the Langhian (Table 3). In the outermost structure studied (Monte Corona), the fractures of set J2 (n=2) yield consistent precipitation temperatures T= 106 ± 8°C and  $\delta^{18}O_{\text{fluids}}$  = 0.0 ±1.8‰ VSMOW; the sample of the set J1 yields a T= 56 ±16°C and  $\delta^{18}O_{\text{fluids}}$  = -1.1 ±1.8‰ VSMOW; in the UMAR, at the Subasio anticline, set F1 (n=3) returns temperatures T ranging from 80 ±5°C to 141 ±19°C and a corresponding  $\delta^{18}O_{\text{fluids}}$  ranging from 8.4 ±1.0‰ to 16.1 ±2.1‰ VSMOW, while the set J2 yields a T=71 ±0°C and  $\delta^{18}O_{\text{fluids}}$  = -5.2 ±0.0‰ VSMOW; in the Monte Nero, set J1 (n=2) yields consistent T = 30 ±15°C and  $\delta^{18}O_{\text{fluids}} = [2.7 \pm 2.4 \text{ to } 6.8 \pm 0.2] \%$  VSMOW; in the syncline on the west of the Monte San Vicino, set F2 (n=2) return T=36 ±4 to 70 ±7°C and  $\delta^{18}$ Ofluids = 2.5 ±0.7 to 8.3 ±1.2‰ VSMOW; in the Monte San Vicino, set J1 yields a T = 47 ±5°C and  $\delta^{18}O_{\text{fluids}}$  = 3 ±1.1‰ VSMOW while set J2 yields a T= 74 ±10°C and  $\delta^{18}O_{\text{fluids}}$ = 7.2 ±1.6‰ VSMOW. In the Monte Catria, the sample of set J1 was characterized by a fluid with  $\delta^{18}O_{\text{fluids}}$  = 8.6 ±0.7‰VSMOW precipitated at T=44 ±4°C, the sample of set J2 by a fluid with  $\delta^{18}O_{\text{fluids}}$  = 11.1 ±2.3‰VSMOW precipitated at T=59 ±10°C. In the easternmost structure (Monte Conero), the sample of J1 was characterized by a fluid with  $\delta^{18}O_{\text{fluids}}$  = 5.8 ±2.4‰VSMOW precipitated at T=42 ±12°C.

f. U-Pb absolute dating of veins and faults

All samples from veins, whatever the set they belong to, reveal to have a U/Pb ratio not high enough to return an age, with a too low U content and/or dominated by common lead (see Supplementary material), which seems to be common in tectonic veins (Roberts et al., 2020). Of the 35 samples screened, 2 faults showed favorable 238U/206Pb ratios to allow for U-Pb dating (FAB5 and FAB6) by applying the mapping approach. In sample FAB5, the pixels with higher U/Pb ratios made it possible to obtain identical ages within the limits of uncertainty for the different plots in spite of a majority of pixel values dominated by common lead ( $5.03 \pm 1.2$  Ma,  $4.92 \pm 1.3$  Ma and  $5.28 \pm 0.95$  Ma for the TW, the 86TW and the isochron plot, respectively) (Fig 9a). The rather large age uncertainties are consistent with the moderately high RSE values, but the d-MSWD values close to 1 indicate good alignment of discretized data (Fig. 9b). In sample FAB6, the mapping approach returned distinct ages according to the plot considered because of low U/Pb ratios (from 2.17  $\pm$  1.4 Ma to  $6.53 \pm 2$  Ma). Keeping in mind their low reliability, the ages obtained for this sample grossly point toward precipitation younger than ~8 Ma.

**4. Interpretation**

a. Sequence of fracturing events and related regional compressional and extensional trends

The previously defined joint/vein, fault and stylolite sets were compared and gathered in order to reconstruct the deformation history at the scale of the belt. We interpret the mesostructural network as resulting from three stages of regional deformation, supported by already published fold-scale fracture sequence ((Tavani and Cifelli, 2010; Tavani et al., 2008; Petracchini et al., 2012; Beaudoin et al., 2016; Díaz General et al., 2015; Di Naccio et al., 2005; Vignaroli et al., 2013), and in line with the ones observed in most recent studies (see Evans and Fischer, 2012; Tavani et al., 2015a for reviews) *: Layer parallel shortening (LPS)* stage: chronological relationships statistically suggest that set J1 formed before set J2. Set J1 is kinematically consistent with set S1 that recorded bedding-parallel, NE-SW directed Apenninic contraction, except in some places where sets J1 and S1 rather formed under a slight local rotation/perturbations of the NE-SW directed compression as a result of structural inheritance and/or of the arcuate shape of the fold. Bedding-parallel reverse faults of set F1 also belong to this LPS stage as they are likely to develop at an early stage of fold growth (Tavani et al., 2015).

*Folding* stage: set J2 reflects local extension perpendicular to fold axis and associated with strata curvature at fold hinges. The extensional trend, i.e. the trend of J2 joints/veins, changes as a function of curvature of fold axes in map view. We also interpret the stylolite peaks of which orientation are intermediate between set S1 and S2 (Fig. 4) as related to the folding stage (Roure et al., 2005).

Late stage fold tightening (LSFT): some tectonic stylolites with horizontal peaks striking NE-SW (set S2) and some veins/joints (set J3) postdate strata tilting and are consistent with late folding strike-slip and reverse faults (set F2). All these mesostructures formed slightly after the fold has locked, still under a NE-SW contractional trend which is now oriented at a high angle to bedding. They mark a late stage of fold tightening, when shortening is no longer accommodated by e.g., limb rotation.

b. Burial depth evolution and timing of contractional deformation

Stylolite roughness inversion applied to bedding-parallel stylolites (BPS) provides access to the maximum depth experienced by the strata at the time vertical shortening was prevailing over horizontal shortening ( $\sigma_1$  vertical) (Ebner et al., 2009b; Koehn et al., 2007; Beaudoin et al., 2019; Beaudoin et al., 2016; Beaudoin and Lacombe, 2018; Beaudoin et al., 2020; Rolland et al., 2014; Bertotti et al., 2017). In this study, we will compare the depth range returned by the inversion of a population of BPS to a local burial model (Fig. 10) reconstructed from the strata thickness documented in wells located in the western-central part of the UMAR (Nero-Catria area) (Centamore et al., 1979; Tavani et al., 2008). The resulting burial curves were constructed from the present-day strata thicknesses corrected from (1) chemical compaction by increasing the thickness by an estimated 18%, then from (2) physical compaction by using the opensource software backstrip (PetroMehas), considering initial porosity of 70% for the carbonates and 40% for the sandstones, and compaction coefficients of 0.58 and 0.30 derived from exponential decrease of porosity with increasing burial for the carbonates and sandstones, respectively (Watts, 2001). The timing of exhumation was further constrained by published paleogeothermometric studies and by the sedimentary record (Caricchi et al., 2014; Mazzoli et al., 2002). To the west, tectonic reconstructions and organic matter paleothermometry applied to the Tuscan Nappe (Caricchi et al., 2014) revealed that most of this unit locally underwent more burial because it was underthrusted below the Ligurian Nappe, but that the western front of the Ligurian Nappe did not reach Monte Corona (Caricchi et al., 2014). We therefore consider a unique burial curve for the whole western UMAR, and we project the range of depth values at which individual BPS stopped being active on the burial curves of the formations hosting the BPS. Recent application of this technique, coupled with absolute dating of vein cements (Beaudoin et al., 2018), showed that the greatest depth that a population of BPS recorded was reached nearly at the time corresponding to the age of the oldest LPS-related veins, suggesting that it is possible to constrain the timing at which horizontal principal stress overcame the vertical principal stress, switching from burial-related stress regime ( $\sigma$ 1 vertical) to LPS ( $\sigma$ 1 horizontal) (Beaudoin et al., 2020). In the case of the UMAR, 800m is the minimum depth at which dissolution stopped along BPS planes, regardless of studied formations. That confirms that burial-related pressure solution (i.e., chemical vertical compaction) likely initiated at depth shallower than 800 m (Ebner et al., 2009b; Rolland et al., 2014; Beaudoin et al., 2019; Beaudoin et al., 2020).

Figure 10 also shows that BPS were active mainly from the Cretaceous (age of deposition of the platform) until Langhian times (~15 Ma), which suggests that  $\sigma$ 1 likely switched from vertical to horizontal at ca. 15 Ma. For the sake of simplicity, we will consider this age of 15 Ma for the onset of LPS, but one must keep in mind that taking into account a 12% uncertainty on the magnitude of the maximum vertical stress derived from stylolite roughness inversion, hence ± 12% on the determined depth, yields a 19 to 12 Ma possible time span for the onset of LPS (from Burdigalian to Serravalian) (Fig. 10). Sedimentary record pins the beginning of folding of the UMAR to the Tortonian (11-7.3 Ma) in the west and to the Messinian (7.3-5.3 Ma) in the east (onshore) (Calamita et al., 1994). Consequently, in the central and western part of the UMAR, we propose that the LPS stage of Apennine contraction lasted about ~7 Ma (from 15 to 8 Ma - Langhian to Tortonian-) before folding occurred. Absolute dating of faults related to late stage fold tightening in the central part of the UMAR further indicates that fold development was over by ~5 Ma, i.e. by the beginning of the Pliocene (5.3-1.75 Ma). We can therefore estimate an average duration of folding in the western-central part of the UMAR is mid-Pliocene (~3 Ma) (Barchi, 2010), we can also estimate the duration of the LSFT to ~2 Ma. In total, the

probable period when the compressive horizontal principal stress  $\sigma$ 1 was higher in magnitude than the vertical stress (*i.e.* until post-orogenic extension) lasted for 9 Ma in the Western-Central part of the UMAR. We propose an average duration of fold growth about 3 Ma, quite in accordance with previous attempts to constrain fold growth duration elsewhere using either syntectonic sedimentation (3 to 10 Ma (Holl and Anastasio, 1993; Anastasio et al., 2018), up to 24 Ma with quiescent periods in between growth pulses (Masaferro et al., 2002)); or mechanical or kinematic modeling applied to natural cases (1 Ma to 8 Ma (Suppe et al., 1992; Yamato et al., 2011)). The combination of bedding-parallel stylolite inversion, burial models and U-Pb dating of vein cements/fault coatings yields a valuable insight into the timing of the different stages of contraction in a fold-and-thrust belt (Beaudoin et al., 2018).

**c. Paleofluid system evolution**

The combined use of BPS inversion and burial curves constrains the absolute timing of LPS, folding and LSFT in the UMAR (Fig. 10). The further combination of this calendar with the knowledge of the past geothermal gradient as reconstructed from organic matter studies in the eastern part of the TN (23°C/km, Caricchi et al., 2014) therefore yields the expected temperature within the various strata at the time vein sets J1, J2 and J3 and faults F1 and F2 formed. Then it is possible to identify whether fluids precipitated at thermal equilibrium or not during the Apenninic contraction. Overall, most calcite grains from vein cements show thin twins (thickness < 5  $\mu$ m) and rectilinear, suggesting deformation at temperature below 170°C (Ferrill et al., 2004; Lacombe, 2010), in line with the maximum expected temperature reached by the Upper Triassic – Eocene carbonate reservoir (120°C, Fig. 10). The fact  $\delta^{13}$ C values of veins are very close to the  $\delta^{13}$ C values of the surrounding host rocks (Fig. 6c) while the  $\delta^{18}$ O values of the veins are different from the  $\delta^{18}$ O values of the host (Fig. 6d) discards that the fluids' original isotopic signatures were lost due to rock buffering. The similarity of  $\delta^{13}$ C values of veins compared to local host-rock, along with the  $^{87}$ Sr/ $^{86}$ Sr values of the veins, that are in accordance with the expected values of the host rocks (Fig. 7) (McArthur et al., 2001), points towards very limited exchange between stratigraphic reservoirs and rules out external fluid input into the system. Indeed, other potential fluid reservoirs such as lower Triassic evaporites seawater have 87Sr/86Sr values significantly higher (0.70800-0.70820) than the highest 87Sr/86Sr values documented in the UMAR (Monte Subasio, 0.70760-0.70770; Fig. 7). These characteristics indicate a closed fluid system in most of the UMAR, with formational fluids precipitating at thermal equilibrium, limited reservoir fluid - host rock interactions in the reservoirs and limited cross-strata fluid migrations.

When considering  $\delta^{18}$ O and  $\Delta_{47}$ CO2 values, the folds in the westernmost part of the UMAR, Monte Corona and Monte Subasio, require a different interpretation from the other folds of the UMAR (Fig. 6d). Indeed, if considering an environment with limited connection between reservoirs and no implication of external fluids, *i.e.* where fluids are sourced locally from the marine carbonates, the fact that in the western part of the UMAR, the  $\delta^{18}$ O values of the veins are significantly more positive than the  $\delta^{18}$ O values of their host-rocks implies that fluids that precipitated were enriched in 18O isotope relative to the original formational fluid. Such a fractionation is usually interpreted as the result of rock dissolution during fluid migration (Clayton et al., 1966; Hitchon and Friedman, 1969). This is further supported by the  $\delta^{18}$ Ofluids values derived from  $\Delta_{47}$ CO2 measurements, that are higher in the Monte Subasio (8 to 16‰ VSMOW, Fig. 8) than in the rest of the UMAR (from 0 to 8‰ VSMOW, Fig. 8), witnessing a higher degree of reservoir fluid-rock interaction in the western part of the UMAR.

Temperatures of precipitation are consistent with the predicted temperatures of host rocks considering the formation and timing of fracture development in most of the UMAR at all times of deformation (Figs. 6-7-10). This is different in Monte Corona and Monte Subasio where veins precipitated from fluids at higher temperature than the predicted temperatures for the host-rock. In the Maiolica Fm. at Monte Corona, J2 veins returned a temperature > 100°C while the maximum predicted temperature during folding is < 90°C. In the Scaglia Fm. at Monte Subasio, the LPS-related veins (J1) and faults (F1) precipitated from fluids much hotter (105 -140°) than the predicted temperature during LPS (<70°C). The interpretation of an easternward migration of hydrothermal basinal brines with high degree of fluid-rock interaction affecting the western part of the UMAR is further supported by higher the 87Sr/86Sr ratios in Monte Subasio (Fig. 7). A hydrothermal dolomitizing fluid migration event was documented in the southeastern part of the UMAR (Montagna dei Fiori), and interpreted as vertical fluid migration from deeper Jurassic reservoirs (Mozafari et al., 2019; Storti et al., 2018). But the reconstructed fluid temperatures (100°C) and  $\delta^{18}O_{\text{fluids}}$ (6‰ VSMOW) are still much lower than the ones reconstructed from the fluids involved in the Monte Subasio (105-140°C; 9 to 15‰ VSMOW), supporting that a different fluid system was prevailing in the westernmost part of the belt during LPS, folding and LSFT compared to central-eastern UMAR.

**5. Discussion**

a. Fluid origin and engine of migration in the westernmost Umbria-Marches Apennine Ridge

During LPS and folding, a temperature of fluid precipitation of up to 140°C, i.e., significantly higher than the local host-rock temperature, implies that fluids flowed from depth >4 km, while a high  $\delta^{18}O_{\text{fluids}}$  reflects a high degree of reservoir fluid-rock interaction. Considering the subsurface geometry of the UMAR (Fig. 1) and discarding any input of external fluids originated from the lower Triassic formations or from the basement on the basis of  ${}^{87}\text{Sr} / {}^{86}\text{Sr}$  values, we propose that the fluids originated from the fluids originated from the westward lateral extension of the carbonate platform reservoir that was buried under the Tuscan Nappe and Ligurian Nappe (Caricchi et al., 2014; Carboni et al., 2020). The coexistence inside a single deformation stage (LPS or folding) of both local/meteoric fluids and hydrothermal brines that migrated from depths can be explained by transient flushes into the system of hydrothermal fluids flowing from deeply buried parts of the same, stratigraphically continuous, reservoir (Bachu, 1995; Garven, 1995; Machel and Cavell, 1999; Oliver, 1986).

We therefore propose that the fluid system prevailing at the Monte Corona and at the Monte Subasio reflects an eastward, squeegee-type, flow of hydrothermal fluids (Fig. 11). The long-term migration engine is the lateral variation of the burial depth of the reservoir related to the stacked Tuscan and Ligurian Nappes in the west of the UMAR (up to 4 km for the Scaglia Fm., Caricchi et al., 2014), that does not affect the UMAR (burial up to 3 km for the Scaglia Fm., Figs. 10, 11b). This depth variation likely created a difference in hydraulic head, and so a 
[revised manuscript text omitted]

**Author contribution**

NB, OL, DK, A. Billi, JPC were involved in the overall writing of the manuscript led by NB; NB, OL, DK, A. Billi collected structural data and rock samples in the field; NB, AL and OL conducted microstructural inversion; GH, A. Boyce, CJ, MM, NR, IM, FC and CP designed experiments and collected the geochemical data and wrote the related parts of the manuscript and appendices. All authors critically reviewed the multiple drafts of the manuscript.

**7. Acknowledgments**

This work has received funding from the Natural Environmental Research Council under grant number IP-1494-1114, from European Union's Seventh Framework Program for research, technological development, and demonstration under grant agreement 316889, and it also received funds from Sorbonne Université (research agreement C14313). NB is funded through the ISITE program E2S, supported by ANR PIA and Région Nouvelle-Aquitaine. Authors also thank two anonymous reviewers for insightful comments that improved the manuscript, and Randolph Williams for his editorial work.

[revised manuscript text omitted]

Figures with captions